# Gene and protein expression and metabolic flux analysis reveals metabolic scaling in liver ex vivo and in vivo

Ngozi D Akingbesote[1,2], Brooks P Leitner[1,2], Daniel G Jovin[1,2], Reina Desrouleaux[1,3], Dennis Owusu[1,2], Wanling Zhu[1,2], Zongyu Li[1,2], Michael N Pollak[4,5], Rachel J Perry[1,2]*

[1]Department of Cellular & Molecular Physiology, Yale University, New Haven, United States; [2]Department of Internal Medicine – Endocrinology, Yale University, New Haven, United States; [3]Department of Comparative Medicine, Yale University, New Haven, United States; [4]Lady Davis Institute for Medical Research, Jewish General Hospital, Montreal, Canada; [5]Department of Oncology, McGill University, Montreal, Canada

*For correspondence:
rachel.perry@yale.edu

Competing interest: The authors declare that no competing interests exist.

**Abstract** Metabolic scaling, the inverse correlation of metabolic rates to body mass, has been appreciated for more than 80 years. Studies of metabolic scaling have largely been restricted to mathematical modeling of caloric intake and oxygen consumption, and mostly rely on computational modeling. The possibility that other metabolic processes scale with body size has not been comprehensively studied. To address this gap in knowledge, we employed a systems approach including transcriptomics, proteomics, and measurement of in vitro and in vivo metabolic fluxes. Gene expression in livers of five species spanning a 30,000-fold range in mass revealed differential expression according to body mass of genes related to cytosolic and mitochondrial metabolic processes, and to detoxication of oxidative damage. To determine whether flux through key metabolic pathways is ordered inversely to body size, we applied stable isotope tracer methodology to study multiple cellular compartments, tissues, and species. Comparing C57BL/6 J mice with Sprague-Dawley rats, we demonstrate that while ordering of metabolic fluxes is not observed in in vitro cell-autonomous settings, it is present in liver slices and in vivo. Together, these data reveal that metabolic scaling extends beyond oxygen consumption to other aspects of metabolism, and is regulated at the level of gene and protein expression, enzyme activity, and substrate supply.

## Editor's evaluation

Key metabolic processes have been shown to scale inversely with the body mass of different animals. This study provides direct evidence for metabolic scaling of key metabolic fluxes in the livers of mice and rats, as well as species-specific differences in the transcription and expression of enzymes involved in energy metabolism that could contribute to metabolic scaling. The finding suggests that metabolic scaling likely reflects multiple levels of regulation and have broad implications for studying animal metabolism and physiology.

## Introduction

In 1932, Max Kleiber published a seminal study (*Kleiber, 1932*), integrating prior reports demonstrating a phenomenon that came to be termed 'Kleiber's law,' or the principle of metabolic scaling. Metabolic scaling refers to the phenomenon that the metabolic processes in many animals, if not all,

scale inversely to three-quarters of their body mass (*West et al., 1997*). In simpler terms, there is a reduction in metabolic rate as body size increases. For example, an elephant is 25 million times larger than a fruit fly, yet its energy expenditure is only 20 thousand times higher; thus, from the fruit fly to elephant, the metabolic rate per gram of body weight scales down 1250 times. While there is experimental evidence for metabolic scaling from bacteria to large mammals, data have been generated almost exclusively from observations of caloric intake and oxygen consumption, with gene and protein expression, and substrate fluxes almost entirely unexplored.

The concept of hierarchical regulation, whereby gene expression initiates the cascade that allows for the flux of metabolic pathways (*Rossell et al., 2005*; *Suarez and Moyes, 2012*), provides a systems framework to begin to understand scaling. Beginning at the transcriptional level, we studied liver gene expression across five species: mice (*Mus musculus*), rats (*Rattus norvegicus*), monkeys (*Macaca mulatta*), humans (*Homo sapiens*), and cattle (*bos taurus*), species with a 30,000-fold range of average body weight in adults (from 30 g in mice, to 900 kg in cattle). Numerous metabolic genes related to glycolysis, gluconeogenesis, fatty acid metabolism, oxygen consumption, electron transport, and redox function, and detoxification of oxidative damage, were expressed at levels inverse to body size. Further analysis of liver proteomics revealed that approximately half of the genes in the liver that were expressed inversely proportionally to body size at the transcriptional level, were also expressed at levels inversely proportional to body size at the level of protein expression. To determine if gene and protein expression would correlate with enzyme activity and metabolic flux, we performed a comprehensive assessment of liver metabolism in vivo and in vitro using modified Positional Isotopomer NMR Tracer Analysis (PINTA) (*Perry et al., 2017b*) and stable isotope-derived turnover (*Perry et al., 2015*) methods. Our analysis shows that rats exhibit lower metabolic rates when compared to mice *in* and *ex* vivo; however, no significant differences were observed when we isolated hepatocytes and cultured them in vitro under identical conditions. Taken together, this study demonstrates the variation of metabolic fluxes according to body size, extending prior studies of metabolic scaling, and provides unique insight into the regulation of metabolic flux across species.

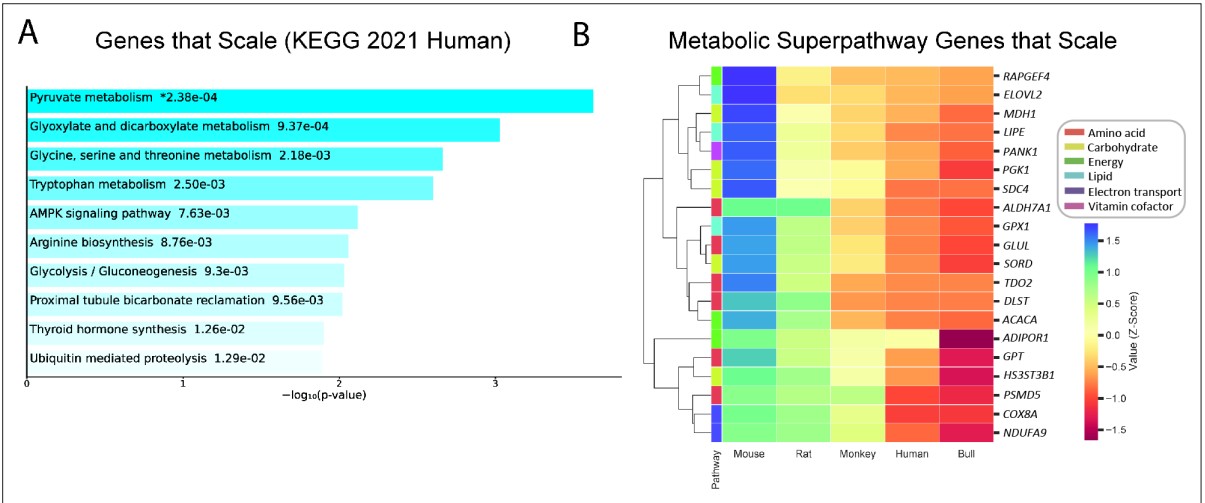

**Figure 1.** Genes that follow the pattern of allometric scaling are most strongly related to metabolism. (**A**) KEGG Pathway enrichment of all genes that are expressed with an inverse correlation to body mass, and (**B**) clustering heatmap of scaled genes that belong to one of six Reactome metabolic superpathways. All samples were obtained from males. For clarity, the human gene (and style of writing human gene names) are shown. *RAPGEF*, rap guanine nucleotide exchange factor; *ELOVL2*, Elongation of Very Long Chain Fatty Acids-Like 2; *MDH1*, malate dehydrogenase 1; *LIPE*, hormone-sensitive lipase E; *PANK1*, pantothenate kinase 1; *PGK1*, phosphoglycerate kinase 1; *SDC4*, syndecan 4; *ALDH7A1*, aldehyde dehydrogenase 7 family member A1; *GPX1*, glutathione peroxidase 1; *GLUL*, glutamate-ammonia ligase; *SORD*, sorbitol dehydrogenase; *TDO2*, tryptophan 2,3-dioxygenase; *DLST*, dihydrolipoamide S-succinyltransferase; *ACACA*, acetyl-CoA carboxylase-alpha; *ADIPOR1*, adiponectin receptor-1; *GPT*, glutamic-pyruvate transaminase; *HS3ST3B1*, heparan sulfate-glucosamine 3-sulfotransferase 3B1; *PSMD5*, proteasome 26 S subunit, non-ATPase-5; *COX8A*, cytochrome c oxidase subunit 8 A; *NDUFA9*, NADH:ubiquinone oxidoreductase subunit A9.

## Results

## Genes within the liver that are expressed inversely proportional to body weight are predominantly metabolic genes

We examined gene expression in livers from mice (*Mus musculus*), rats (*Rattus norvegicus*), monkeys (*Macaca mulatta*), humans (*Homo sapiens*), and cattle (*Bos taurus*). Using recent advances in high throughput mRNA sequencing and bioinformatics tools that allow for intra-species data preprocessing (*Bray et al., 2016*; *Conesa et al., 2016*; *Ritchie et al., 2015*), we searched for a set of genes in the liver, the metabolic hub of mammals, whose expression correlates inversely with body mass. After normalizing for differences in transcript length and abundance across species, we filtered out genes that followed the pattern of mouse >rat > monkey >human > cow. The genes that met these criteria were predominantly related to metabolic pathways, including pyruvate metabolism, amino acid metabolism, and glucose metabolism (*Figure 1A*). Genes from this list were further restricted to genes involved in amino acid, carbohydrate, energy, lipid, vitamin, and TCA cycle metabolism, and

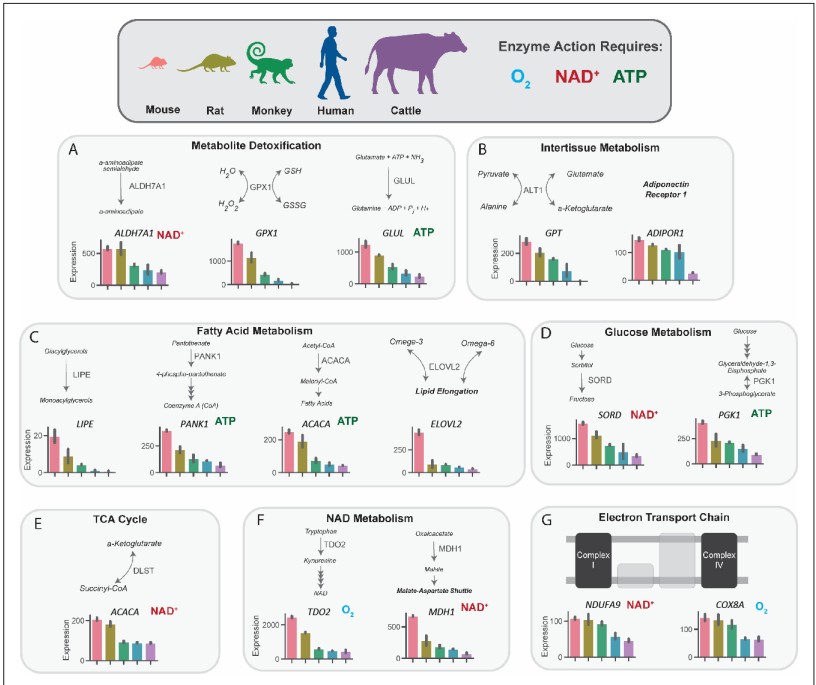

**Figure 2.** Metabolic genes that are expressed inversely proportionally to body size implicate key pathways in substrate and nucleotide supply, glucose and fatty acid flux, oxygen consumption, and detoxification pathways. mRNA expression of key regulatory genes related to metabolite detoxication (**A**), intertissue metabolism (**B**), fatty acid metabolism (**C**), glucose metabolism (**D**), tricarboxylic acid (TCA) cycle, NAD metabolism (**F**), and the electron transport chain (**G**) in mice, rats, monkeys, humans, and cattle. Bars denote expression levels by an organism, following the same order shown in the cartoon of organisms. Expression was normalized to counts per million and was then further normalized for sequencing depth and transcript length. All genes met an adjusted p-value threshold of 0.01 using a one-way ANOVA with the Bonferroni correction for multiple comparisons. All samples were obtained from males (n=2 replicates per species). *ALDH7A1*, aldehyde dehydrogenase 7 family member A1; *GPX1*, glutathione peroxidase 1; *GLUL*, glutamate-ammonia ligase; *GPT*, glutamic-pyruvate transaminase; *ADIPOR1*, adiponectin receptor-1; *LIPE*, hormone-sensitive lipase E; *PANK1*, pantothenate kinase 1; *ACACA*, acetyl-CoA carboxylase-alpha; *ELOVL2*, Elongation of Very Long Chain Fatty Acids-Like 2; *SORD*, sorbitol dehydrogenase; *PGK1*, phosphoglycerate kinase 1; *DLST*, dihydrolipoamide S-succinyltransferase; *TDO2*, tryptophan 2,3-dioxygenase; *MDH1*, malate dehydrogenase 1; *NDUFA9*, NADH:ubiquinone oxidoreductase subunit A9; *COX8A*, cytochrome c oxidase subunit 8 A.

The online version of this article includes the following source data and figure supplement(s) for figure 2:

**Source data 1.** Source data for *Figure 2* and *Figure 2—figure supplement 1*.

**Figure supplement 1.** Liver metabolic genes, but not structural genes, are expressed inversely proportionally to body size in rodents.

demonstrated a range of degrees of inverse correlation with body mass, with only TCA cycle genes clustering together (*Figure 1B*).

## Genes encoding enzymes involved in hepatic metabolism are expressed inversely proportionally to body mass and involve metabolite detoxification, intertissue metabolism, substrate metabolism, electron transport, and NAD metabolism

In order to further understand the functional aspects of the metabolic genes that are expressed inversely proportionally to body size, the gene list from *Figure 1B* was categorized into several functional categories, converging on optimizing energy provision, oxidative metabolism, and damage control from oxidative stress and ammonia (*Figure 2*). Furthermore, to understand whether or not certain genes that are expressed inversely proportionally to body size involved anabolic or catabolic processes, they were further classified by their properties to be energy suppliers or consumers. Eleven of sixteen critical metabolic enzymes that scaled required molecular oxygen, NAD+/NADH, or ATP/ADP for function, possibly indicating exquisite regulation of energy-consuming processes at the individual gene level. Genes involved in the detoxication of lipid peroxidation-derived aldehydes (*ALDH7A1*), hydrogen peroxide (*GPX1*), and ammonia (*GLUL*) suggest scaling of damage control mechanisms that are associated with increased oxidative metabolism across species (*Figure 2A*). The inverse correlation between body size and expression of genes that are associated with interorgan crosstalk is consistent with scaling in vivo which would not be expected in plated cells. For example, the differentially expressed genes include *GPT1*, which is involved in recycling skeletal muscle-derived alanine back to liver-derived glucose (*Felig and Wahren, 1971*; *Petersen et al., 2019*), and the adiponectin receptor (*ADIPOR1*), which binds an adipose tissue-derived hormone that regulates gluconeogenesis and fatty acid oxidation (*Lin and Accili, 2011*; *Li et al., 2020 Figure 2B*). Genes involved in fatty acid metabolism included the rate-limiting steps of the synthesis of CoA (*PANK1*), of de novo fatty acid synthesis (*ACACA*), and of fatty acid elongation (*ELOVL2*), in addition to the oxidation of diacylglycerols (*LIPE*) (*Figure 2C*). NAD and ATP-dependent genes involved in glycolysis (*PDK1*), fructose/glucose metabolism (*SORD1*), and *DLST* of the TCA cycle also correlated inversely with body size (*Figure 2D–E*). Differentially regulated genes also couple oxygen consumption to NAD provision (*MDH1, TDO2*), and are involved with the function of the electron transport chain (subunits of complex I, *NDUFA9*, and complex IV, *COX8A*, which catalyzes oxygen accepting the final electrons of the electron transport chain) (*Figure 2F–G*).

To examine the possibility that the inverse correlation between body mass and gene expression observed in the transcriptomics analysis could be a consequence of global alterations in mRNA (for example, as a consequence of alterations in RNA turnover rates), we performed targeted quantitative polymerase chain reaction (qPCR), measuring in liver tissue abundance of mRNA encoding several enzymes that were found to scale in the five-species transcriptomics analysis, relative to the common housekeeping gene β-actin (*Actb*). We found that all three enzymes (*Glul*, *Lipe*, and *Dlst*) scaled relative to *Actb* (*Figure 2—figure supplement 1A–C*), whereas structural genes (collagenase 3 [*Mmp3*] and *Larp1*) did not (*Figure 2—figure supplement 1D–E*), indicating that the differences in metabolic gene expression observed across species is likely not a result of global changes in RNA levels.

In addition to transcriptomics, we assessed proteomics data to evaluate the protein levels corresponding to the genes that were found to be expressed inversely proportionally to body size at the level of mRNA expression. Our proteomics data were limited to mouse, rat, and human, as all the open-source proteomic databases that we identified lacked data from monkey or cow. An important limitation for finding such data is that even with careful post-processing, we cannot combine data from different studies, because differences in methods of tissue preparation may influence results. Therefore, we were limited to a single experiment that had generated proteomics data for mouse, rat, and human using the same experimental procedures. The dataset contained protein expression corresponding to eight of the twenty genes identified to scale in our transcriptomics data analysis. Of these, three (GLUL, GPX1, and MDH1) were found to follow a reverse correlation with body size (*Figure 3A–C*). Interestingly, one of these proteins (GLUL) was also found to be expressed inversely proportionally to body size in the left ventricle of the heart (*Figure 3D*). Additionally, we measured liver transaminase concentrations and observed that both alanine aminotransferase (ALT) and aspartate aminotransferase (AST) exhibited lower concentrations in humans as compared to rats and rats

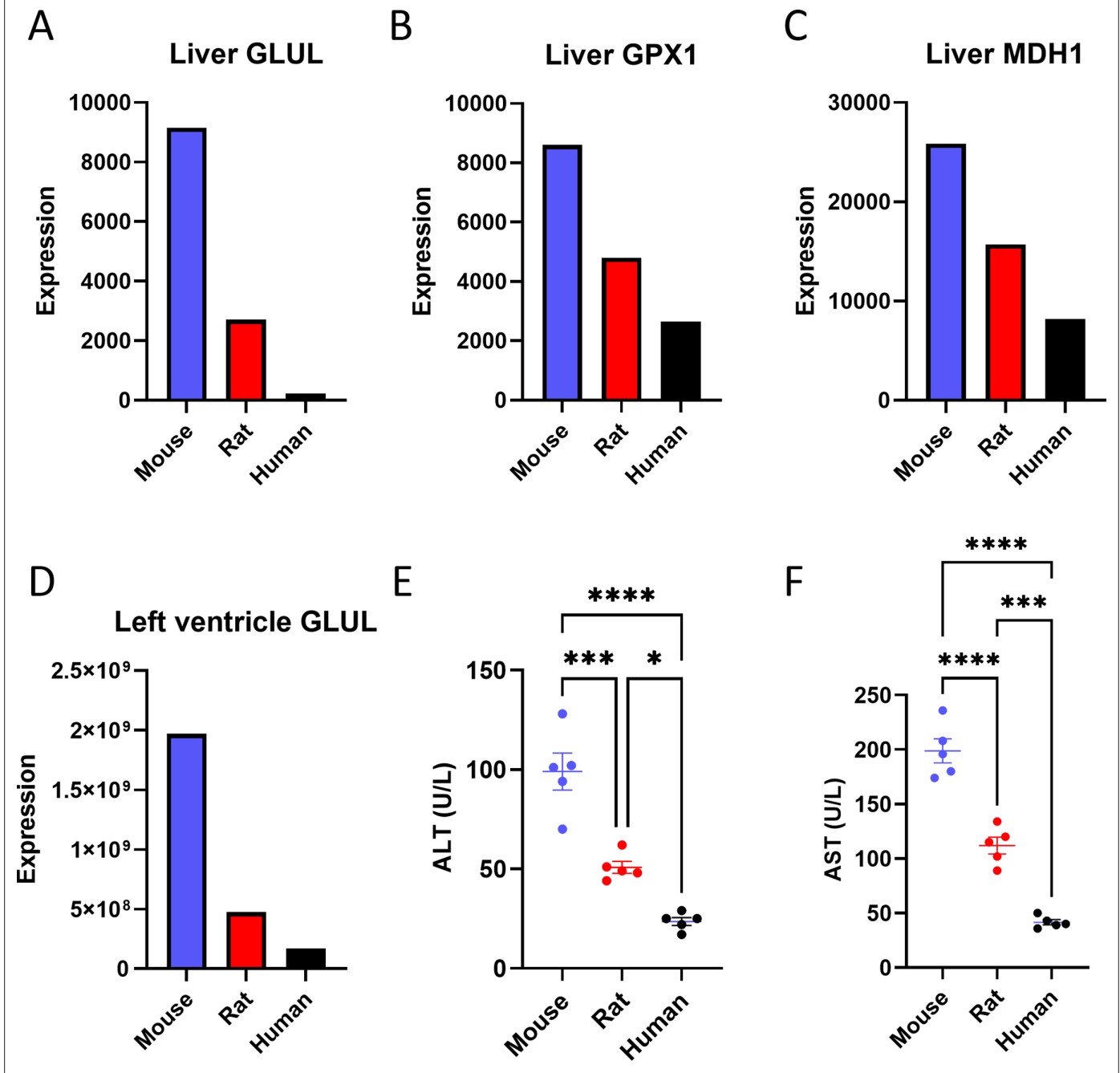

**Figure 3.** Proteomics reveals a negative correlation between body size and the expression of some liver proteins. Liver (**A**) glutamate-ammonia ligase (GLUL), (**B**) glutathione peroxidase 1 (GPX1), and (**C**) malate dehydrogenase 1 (MDH1) protein expression. (**D**) GLUL protein expression in the left ventricle of the heart. The proteomics analysis was performed on n=1 per species, so statistical comparisons were not possible. (**E**) Plasma alanine aminotransferase (ALT) and (**F**) aspartate aminotransferase (AST) concentrations (for both transaminases, n=5 per species). *p<0.05, ***p<0.001, ****p<0.0001.

The online version of this article includes the following source data and figure supplement(s) for figure 3:

**Source data 1.** Source data for *Figure 3* and *Figure 3—figure supplement 1*.

**Figure supplement 1.** Liver metabolic enzyme activity is inversely proportional to body size.

as compared to mice (*Figure 3E–F*), consistent with scaling at the level of protein expression as well as mRNA expression. Finally, we utilized established enzymatic assays to measure the activity of peroxidase and pyruvate carboxylase in the livers of mice and rats. 30–40% lower activity of each enzyme per mg tissue was observed in rats as compared to mice (*Figure 3—figure supplement 1A–B*), suggesting scaling at the level of metabolic enzyme activity.

## Metabolic rates of mouse vs. rat hepatocytes in vitro are not significantly different

Considering prior data reporting higher oxygen consumption per unit body mass in smaller as compared to larger animals (*Gilman et al., 2013*; *Brody, 1945*; *Urbina and Glover, 2013*), we first asked whether these differences were cell-intrinsic, or whether in vivo or hepatocyte-extrinsic signals are required. We incubated plated hepatocytes in [3-$^{13}$C] lactate and first validated that the data met the assumptions of PINTA, including reaching steady-state in [$^{13}$C] lactate and glucose enrichment, and producing glucose at a linear rate throughout the 6 hr incubation (*Figure 4—figure supplement 1A–C*). Consistent with the possibility that hepatocyte-extrinsic signals are primarily responsible for metabolic scaling, when we used PINTA to assess cytosolic and mitochondrial fluxes, we observed no significant differences between species in any of the fluxes measured in plated hepatocytes: glucose production, $V_{PC}$, $V_{CS}$, the contribution of glucose or fatty acids to the tricarboxylic acid (TCA) cycle, or lipolysis (*Figure 4A–H*, *Figure 4—figure supplement 1D–F*). Similarly, a mitochondrial stress test in plated hepatocytes revealed no difference in any parameter: neither basal mitochondrial and non-mitochondrial respiration, ATP production, maximal (uncoupled) respiration, spare respiratory capacity, nor proton leak differed between plated hepatocytes from mice and rats (*Figure 4I*). Previous studies have demonstrated scaling in vitro in cell suspensions only when analyzed immediately after hepatocyte isolation (*Porter and Brand, 1995*), and have suggested that the phenomenon of scaling gradually disappears around 24 hr post removal (*Brown et al., 2007*), similar to the conditions in which we performed these studies. Most prior in vitro studies have also demonstrated an absence of scaling, in contrast to in vivo (*Glazier, 2015*), and we extend these results to gluconeogenic and lipolytic fluxes in hepatocytes, glucose production in liver slices, and multimodal flux analysis in vivo.

## Glucose production per gram tissue is higher ex vivo in liver slices from mice than in rats

Next, considering that hepatocytes comprise approximately 70–80% of liver mass and that their culture in vitro does not replicate in vivo conditions (*Krebs, 1950*), we asked whether glucose production would be different between mice and rats in slices of liver. Indeed, we found that liver glucose production per gram liver mass was threefold greater in mouse liver slices as compared to rats (*Figure 5A–B*), suggesting that hepatocyte-extrinsic signals (for example, from other liver cell types) are involved in liver metabolic scaling.

## Metabolic rates in multiple tissue types are higher in vivo in mice relative to rats

We utilized multimodal stable isotope metabolic flux analysis to compare rats and mice with respect to a panel of metabolic fluxes (*Figure 6A*). First, we validated tracer assumptions in vivo, including the metabolic and isotopic steady state in plasma and negligible liver glycogen concentrations, although in the recently fed state, hepatic glycogenolysis was higher in mice than that in rats (*Figure 6—figure supplement 1A–G*). Using PINTA, we found that both endogenous glucose production and gluconeogenesis from pyruvate ($V_{PC}$) per gram liver were more than twofold higher in mice than rats (*Figure 6B–C*), although the fractional contribution of pyruvate to gluconeogenesis did not differ between mice and rats (*Figure 6—figure supplement 1H*). Mitochondrial oxidation scaled similarly, increasing threefold in mice as compared to rats studied under the same conditions, due to increases in both glucose oxidation (pyruvate dehydrogenase flux, $V_{PDH}$) and fatty acid oxidation (*Figure 6D–F*), associated with an increase in the ratio of pyruvate carboxylase anaplerosis to citrate synthase flux ($V_{PC}/V_{CS}$) without any difference in the fraction of $V_{CS}$ flux fueled by glucose through PDH (*Figure 6—figure supplement 1I–J*). While we did not have the capacity to measure liver fluxes in larger mammals in the current study, endogenous glucose production, $V_{PC}$, and $V_{CS}$ previously measured using PINTA were 50–60% lower in overnight fasted humans than in rats (*Petersen et al., 2019*), assuming a liver

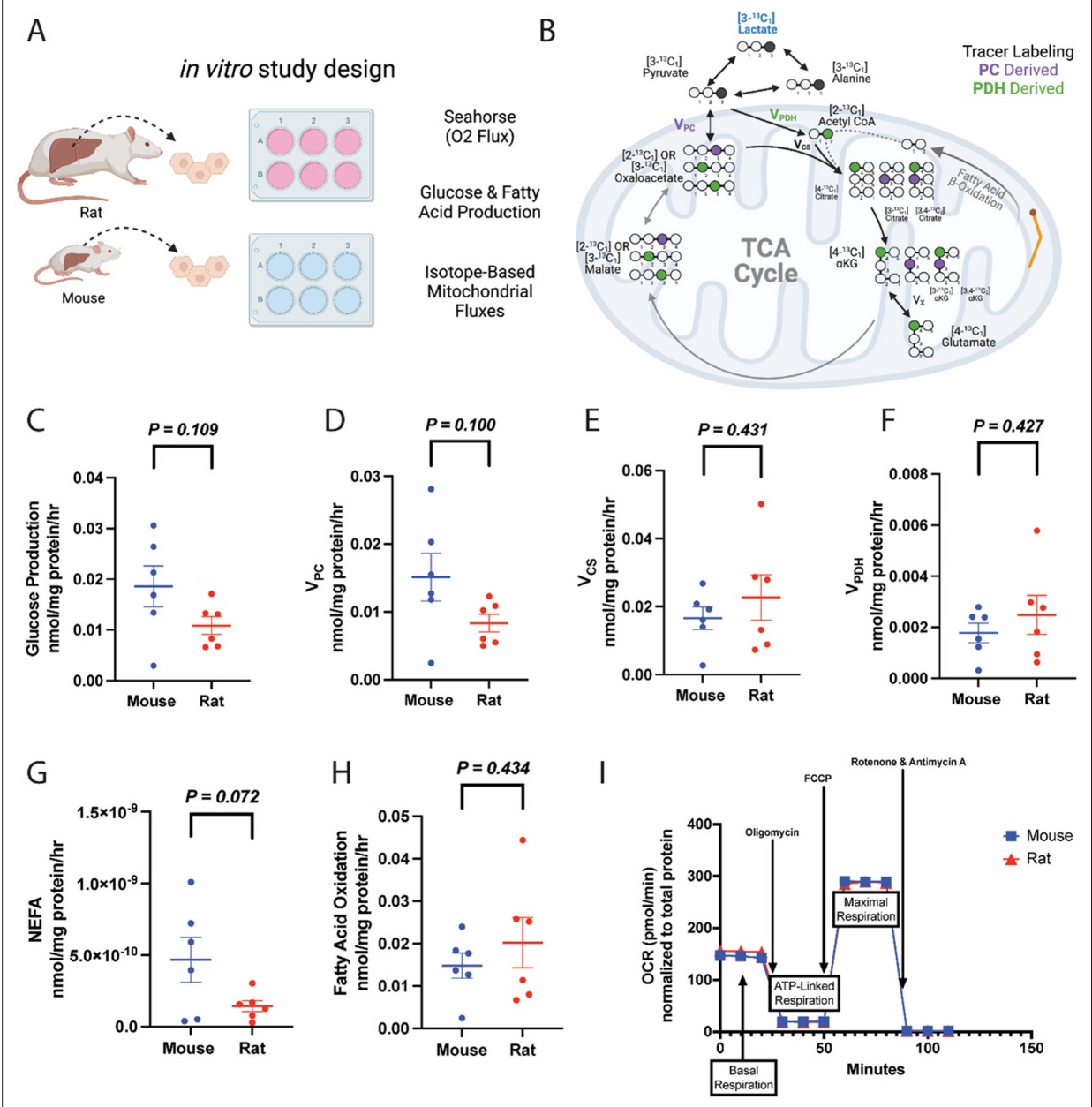

**Figure 4.** Metabolic fluxes are not different between mouse and rat hepatocytes in vitro. (**A**) Study design. This figure was made using Biorender.com. (**B**) Tracer labeling strategy. (**C**) Glucose production. (**D**) Gluconeogenesis from pyruvate (pyruvate carboxylase flux, $V_{PC}$). (**E**) Citrate synthase flux ($V_{CS}$), i.e., mitochondrial oxidation. (**F**) Pyruvate dehydrogenase flux ($V_{PDH}$), i.e., the contribution of glucose via glycolysis to total mitochondrial oxidation. (**G**) Non-esterified fatty acid (NEFA) production. (**H**) The contribution of fatty acid oxidation to citrate synthase flux. (**I**) Oxygen consumption rate (OCR) during a mitochondrial stress test. In all panels, hepatocytes from wild-type males were studied, and groups were compared using the two-tailed unpaired Student's t-test. No significant differences were observed. In all panels, the mean ± SEM. of six biological replicates (averaged from three technical replicates per biological replicate) is shown.

The online version of this article includes the following source data and figure supplement(s) for figure 4:

**Source data 1.** Source data for *Figure 4* and *Figure 4—figure supplement 1*.

**Figure supplement 1.** Validation of tracer assumptions and flux ratios in the in vitro hepatocyte PINTA flux studies.

size of 1500 g in humans. These differences in metabolic fluxes according to body size applied not only to rodent liver metabolism but also to adipose tissue metabolism: whole-body fatty acid turnover, reflecting lipolysis, was 2.5-fold higher in mice than in rats (***Figure 6G***). No sex differences were observed in any of the measured fluxes (***Figure 6—figure supplement 1K–P***). Taken together, these

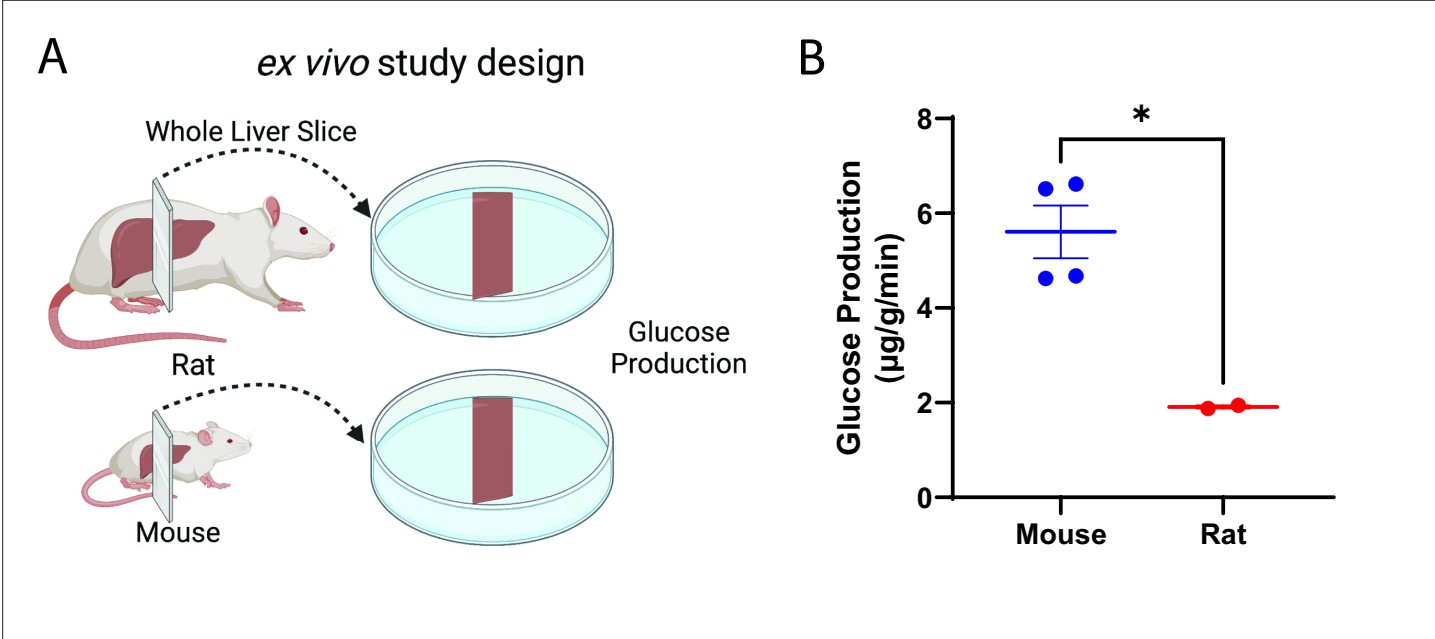

**Figure 5.** Glucose production scales ex vivo in liver slices. (**A**) Study design. This figure was made using Biorender.com. (**B**) Glucose production. Groups were compared by the two-tailed unpaired Student's t-test. Liver slices from male, wild-type animals (n=4 mice and 2 rats, three technical replicates per biological replicate) were studied.

The online version of this article includes the following source data for figure 5:

**Source data 1.** Source data for *Figure 5*.

data emphasize the inadequacy of common in vitro methods as a readout of in vivo metabolism: whereas in vivo mitochondrial oxidation (TCA cycle flux) was threefold higher in mice than in rats, in vitro measurements of oxygen consumption throughout a mitochondrial stress test, TCA cycle flux, and glucose production were not different between the species (*Figure 4*).

## Classical clustering defined species-specific clusters based on in vivo metabolic fluxes but not in vitro fluxes

A clustering dendrogram was applied to our in vitro flux data and showed no distinct clustering between species (*Figure 7A–B*). However, the in vivo metabolic flux data led to distinct clustering of rats and mice (*Figure 7C*), providing a classical clustering-based objective analysis of in vitro versus in vivo metabolic flux.

## Discussion

Oxygen consumption has been shown to scale inversely with body mass in species ranging in mass across 20 orders of magnitude, from $10^{-14}$ to $10^6$ grams (*Ernest et al., 2003*; *Gillooly et al., 2001*; *Kleiber, 1932*; *Makarieva et al., 2005a*; *Makarieva et al., 2008*; *Savage et al., 2004*; *West et al., 2002*). This phenomenon has been most studied in mammals, but is highly conserved, having also been shown to occur in prokaryotes (*Fenchel and Finlay, 1983*; *Makarieva et al., 2005a*; *Makarieva et al., 2005a*; *Makarieva et al., 2008*; *Moses et al., 2008*), plants (*Makarieva et al., 2005b*; *Mori et al., 2010*; *Reich et al., 2006*), insects (*Chown et al., 2007*; *Maino and Kearney, 2014*; *Makarieva et al., 2005a*), fish (*Clarke and Johnston, 1999*; *Gjoni et al., 2020*; *Rubalcaba et al., 2020*), and birds (*Glazier, 2008*; *Hudson et al., 2013*; *Makarieva et al., 2005b*). However, a major limitation of prior studies in this field has been that observations have been largely limited to oxygen consumption and caloric intake, leaving other metabolic processes unexplored. This study sought to address this issue by examining the generalizability of the inverse relationship between body mass and metabolic rates, using both experimental measurements and previously assembled databases that have not previously been employed in this context.

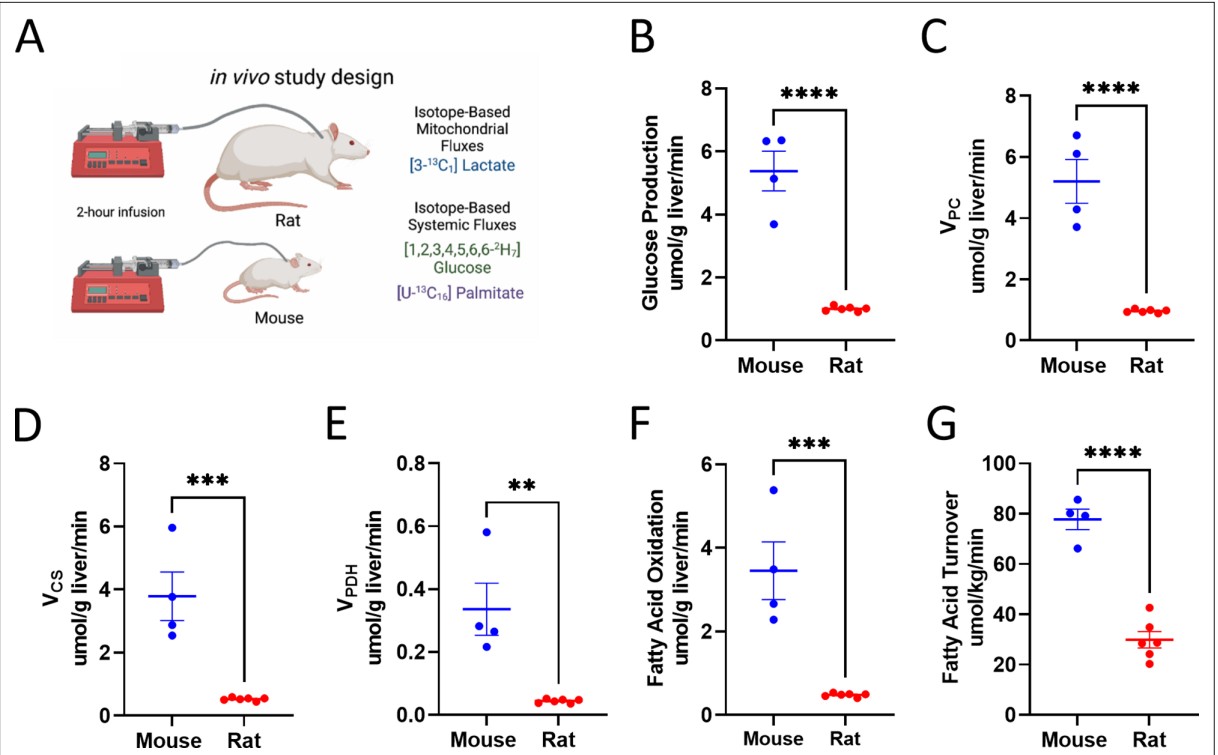

**Figure 6.** Analysis of systemic metabolic fluxes suggests in vivo metabolic scaling in mice vs. rats. (**A**) Study design. (**B**) Endogenous glucose production. (**C**) Gluconeogenesis from pyruvate ($V_{PC}$). (**D**) $V_{CS}$, i.e., mitochondrial oxidation. (**E**) $V_{PDH}$, i.e., the contribution of glucose via glycolysis to total mitochondrial oxidation. (**F**) Palmitate (fatty acid) turnover. (**G**) The contribution of fatty acid oxidation to citrate synthase flux. In all panels, groups were compared using the two-tailed unpaired Student's t-test. Male rodents (n=4 mice and 6 rats) were studied.

The online version of this article includes the following source data and figure supplement(s) for figure 6:

**Source data 1.** Source data for *Figure 6* and *Figure 6—figure supplement 1*.

**Figure supplement 1.** Validation of tracer assumptions, flux ratios, and lack of sex differences in the in vivo flux studies.

It is important to note that the metabolic processes which we observed to be higher in mice as compared to rats did not necessarily adhere quantitatively to the classic metabolic scaling relationship, with metabolic rates proportional to three-quarters of body mass. This speaks to the idea that the scaling relationship is multidimensional: it is entirely conceivable that whole-body oxygen consumption could be proportional to three-quarters of body mass, while other metabolic processes may exhibit a different scaling relationship. Further studies across species beyond rodents will be required to address this question.

The possibility that gene expression, as reflected by mRNA abundance, may also scale with body mass has not been previously addressed. We observed that the expression of key genes in glycolysis, gluconeogenesis, fatty acid metabolism, NAD synthesis and transport, mitochondrial oxygen consumption, and protection from oxidative damage scale with body mass. More compelling, however, is the observation that those genes for which an inverse relationship of expression with body mass is observed, are not randomly distributed across the genome. Rather, the collection of genes whose expression is inversely correlated with body mass is enriched for genes related to metabolic processes, and whose corresponding proteins' enzymatic action are constrained by the supply of substrate, NAD, ATP, or oxygen.

The notion that body mass is a variable related to the level of expression of certain genes has not previously been considered as an aspect of metabolic scaling. However, it should be noted that metabolic scaling cannot fully be explained at the transcriptional level, because many rate-limiting enzymes in the metabolic processes measured in vivo did not scale at the transcriptional level, and only approximately half of genes that scaled at the level of mRNA scaled at the level of protein. Thus, it is likely that both transcriptional and other mechanisms – such as enzyme activity – are responsible

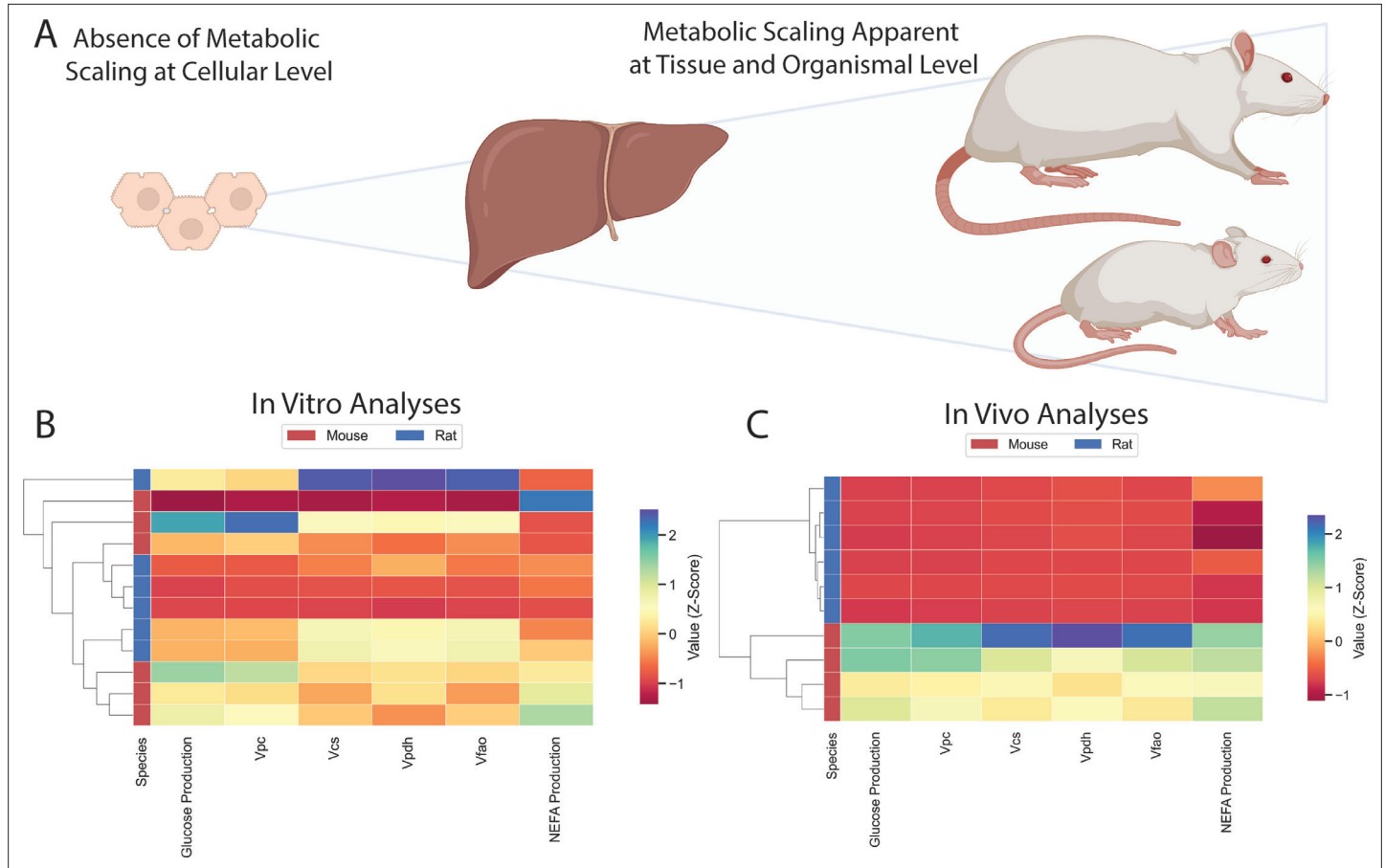

**Figure 7.** Comparison of in vitro and in vivo results. (**A**) Study workflow. (**B**) Clustering heatmap demonstrating the absence of metabolic differences in vitro. (**C**) Clustering heatmap demonstrating metabolic differences between mice and rats in vivo. In panels (**B**) and (**C**), mouse and rat color legends correspond to the species label attached to the dendrogram on the leftmost of each graph. $V_{pc}$ = pyruvate carboxylase flux, $V_{cs}$ = citrate synthase flux, $V_{pdh}$ = pyruvate dehydrogenase flux, $V_{fao}$ = fatty acid oxidation, NEFA = non-esterified fatty acid concentrations. All data presented in *Figures 3 and 5* were utilized in the classical clustering analysis and are included in this figure.

for variations in metabolic flux per unit mass, inversely proportionally to body size. Additionally, the currently available data do not allow us to assess whether the expression of certain isoforms of key metabolic enzymes scales differentially across species.

It is also informative to contrast the lack of an inverse relationship between body size and metabolic fluxes per tissue weight (oxygen consumption, mitochondrial oxidation, lipolysis, and glucose production in hepatocytes) measured in the in vitro setting, to our findings in vivo (where all fluxes proceeded faster in mice than in rats). This emphasizes the need to employ tracer methods in vivo to generate a comprehensive picture of differences in metabolic fluxes between species.

Our findings emphasize that measurement of oxygen consumption in vitro may fail to detect any influence of scaling processes present in vivo. Glucose production was threefold higher in mouse liver slices relative to rat liver slices, but did not significantly differ between plated hepatocytes from mice and rats. Future studies using metabolic flux analysis may have the further capacity to generate new insights as to the scope and mechanism of metabolic scaling (*Wiechert, 2001*). For example, our data do not allow us to ascertain whether differences in oxygen consumption orchestrate metabolic alterations, as has been suggested in the setting of cancer cells (*Nakazawa et al., 2016*), or metabolic alterations require changes in oxygen consumption. Additionally, there are limitations to the fact that metabolic flux studies were performed only in the two arguably most related species included in the transcriptomics analysis (with monkey and human perhaps being similarly related). Our laboratory does not have the capacity to perform flux analysis in larger or smaller species, but we fully recognize that widening our scope beyond the 10-fold range in body size between mice and rats may yield

different results. For example, it is possible that the mathematical relationship between body size and metabolic flux may reach a threshold of body size, after which this relationship may change. However, experimental and computational data suggest that the metabolic scaling relationship with regard to oxygen consumption holds at least from birds to cattle, and that a similar – but not identical – scaling relationship exists in plants (*Niklas and Kutschera, 2015*), so we posit that it is likely that if such a threshold exists, it occurs at a body size orders of magnitude larger than the rodents included in our study.

Taken together, the findings of this study show that the phenomenon of metabolic scaling extends to processes beyond oxygen consumption and caloric intake, and is bookended by scaling not only of gene and protein expression, but also of enzyme activity and cytosolic and mitochondrial fluxes in vivo. It may be that if one were to alter the activity of an enzyme in a larger species using a mutation to increase it to the level of activity of the same enzyme in a smaller species, that the phenomenon of metabolic scaling might disappear. Future work will be required to examine this possibility. The mechanisms underlying the phenomenon of metabolic scaling remain obscure, and deserve investigation as a fundamental question in biology (*Hatton et al., 2019*; *Kolokotrones et al., 2010*; *White and Seymour, 2003*). Our data support the notion that cell-extrinsic regulatory factors are involved.

### Ideas and speculation

These data have implications for metabolic regulation during hibernation, in which metabolic rates are reproducibly and markedly but reversibly suppressed (*Jansen et al., 2021*; *Jansen et al., 2019*; *Tøien et al., 2011*). Metabolic scaling phenomena may also have medical relevance. Recent reports provide data consistent with 'Peto's paradox' by showing that cancer is not more prevalent in larger, long-lived organisms than smaller ones (*Makarieva et al., 2005a*; *Vincze et al., 2022*), despite the fact that more cells are at risk for transformation, and over longer periods of time. Metabolic scaling provides a potential explanation: the slower rates of metabolic processes (including oxygen consumption) and reduced oxidative damage in larger animals imply lower rates of proliferation and carcinogenic DNA damage, so that while more cells are at risk in large animals, carcinogenesis proceeds more rapidly in smaller ones.

Metabolic scaling phenomena that lead to different rates of metabolic fluxes according to organism size deserve consideration in the context of murine models of human diseases. These differences may contribute to the faster rate of aging and shorter life expectancy of rodents as compared to humans. A separate example concerns cancer models, where species-specific different rates of in vivo metabolic fluxes might be relevant to carcinogenesis or efficacy of treatments.

## Materials and methods

All animal studies were approved by the Yale University Institutional Animal Care and Use Committee (protocol 2019–20290). Source data are found in *Supplementary file 1*.

### Rodents

Healthy male (or, when designated, female), wild-type C57BL/6 J mice (catalog number 000664) and Sprague-Dawley rats (strain code 400) were obtained at 8 weeks of age from Jackson Laboratories and Charles River Laboratories, respectively, and given *ad lib* access to regular chow and water. Rodents underwent surgery under isoflurane anesthesia to place catheters in the jugular vein (mice) and in both the jugular vein and carotid artery, with the tip of the arterial catheter advanced into the right atrium of the heart (rats). After a week of recovery and confirmation that the animals had regained their pre-surgical body weight and following a 24 hr fast, rodents underwent the in vivo tracer studies described below. Rodents used for hepatocyte studies were fed *ad lib* until isoflurane euthanasia and liver isolation as described in the in vitro studies below.

### Human participants

Plasma samples were obtained, deidentified, from healthy, lean participants (all male, in order to maintain consistency with the rodent studies; age 26±3, body mass index 24.2 ± 0.8; ethnicity data not collected) in a Yale University IRB-approved (protocol 0108012609, NCT03738852, beginning 11/7/2018) clinical study. Eligibility included age >18 years and body mass index >18.0, without

evidence of renal, thyroid, neurologic, or psychiatric dysfunction. Pre-defined outcomes were focused on the counterregulatory response to hypoglycemia, but baseline samples could be used in a deidentified manner for other analyses. All volunteers provided informed consent to participate in the study, and for the data to be published. Data were collected at Yale University. Blood was obtained from a venous catheter in the left forearm, centrifuged, and stored at –80 °C to await further analysis. No harms were observed.

## Gene expression analysis

All liver raw transcriptomics data were obtained from Array Express (*Brazma et al., 2003*) (https://doi.org/10.1093/nar/gkg091), and were all preprocessed using the same methods. Cattle, monkey, and rat were obtained from E-MTAB-4550, mouse from E-MTAB-5166, and human from E-MTAB-6814. Two replicates from each species were used. Raw counts from each species were normalized to counts per million (CPM) and were then TMM-normalized to account for differences in sequencing depth as well as transcript length across species and scanners using the R package edgeR (*Robinson et al., 2010*). All transcript homologs were converted to human gene names using ENSEMBL in the biomaRt R package (*Durinck et al., 2009*). For the analyses in *Figure 5*, genes were filtered to those that followed the allometric scaling pattern mouse > rat > monkey > human > cattle. This gene list was put into EnrichR and KEGG pathway enrichment analysis was performed (*Kuleshov et al., 2016*). Genes in this list that followed the scaling pattern were then filtered based on whether they met one of six Reactome metabolic superpathways related to the metabolism of amino acids, carbohydrates, energy, lipids, tricarboxylic acid cycle, and vitamin cofactors (*Jassal et al., 2020*). A clustering heatmap was performed on these genes using the Seaborn Python package (*Waskom, 2021*). Genes were plotted again upon this map using the Seaborn Python package. To assess for differences in gene expression in genes displayed on the metabolic map, a one-way ANOVA with a Bonferroni correction for multiple comparisons (thus we used an adjusted p-value threshold for this test of 0.01, corresponding to an alpha level of 0.05, for significance) using the Python Scipy package (*Virtanen et al., 2020*).

Targeted gene expression analysis was performed by qPCR in our laboratory. mRNA was prepared using TRIzol (Life Technologies), purified using RNeasy mini-columns (Qiagen), and used to prepare cDNA. Quantitative real-time PCR was performed using SYBR Green (Applied Biosystems). Primers were obtained from IDT with the sequences shown in *Supplementary file 1*.

## Protein expression analysis

All liver raw mass spectrometry proteomics data were obtained from the Proteomics Identifications Database (PRIDE). Mouse liver S9 fraction (PXD000733), rat liver S9 fraction (PXD000717), and human liver S9 fraction (PXD00734) raw data from a previously published study (*Golizeh et al., 2015*) were processed using MaxQuant software (version 2.1.4.0, Max-Planck Institute of Biochemistry, Munich, Germany), and proteins were identified with built-in Andromeda search engine based on Uniprot canonical protein collections for each species. The false discovery rate cut-offs were set to 1% on peptide, protein, and site decoy levels (default), thereby only allowing the high-quality identifications to pass. Raw intensities of all species showed similar distributions; therefore, data were normalized across species by median centering, and we used the EggNOG database to match orthologous proteins between species. Perseus was used to perform data analysis. We looked exclusively at the twenty genes that were identified with the pattern mouse > rat > human in our transcriptomics data analysis. We used iBAQ intensity (intensity Based Absolute Quantification) as a proxy for protein expression because iBAQ intensities are the raw intensities divided by the number of theoretical peptides. Therefore, iBAQ values are proportional to the molar quantities of the proteins. Additionally, the iBAQ algorithm can roughly estimate the relative abundance of proteins within a sample (*Krey et al., 2018*).

Because the primary goal of *Golizeh et al., 2015* , was to evaluate protein/peptide-level ion exchange fractionation and proteome coverage in liver microsomes versus S9 fractions, the protein data did not include protein abundance. Consequently, the dataset did not have replicates as two livers from each species were digested differently (one with trypsin and the second with pepsin). Since the different digestion methods could alter both protein identification and abundance, we evaluated only the data from the liver S9 fraction digested with trypsin. Since we had no replicates, statistical analysis could not be carried out.

## In vitro tracer analysis

Primary hepatocytes were isolated by the Yale Liver Center's Cell Isolation Core, plated in a six-well plate ($4.0 \times 10^5$ cells per well), and allowed to recover for 6 hr at 37 °C in substrate replete media (DMEM high glucose containing 10% FBS, 2% penicillin–streptomycin, 100 nM dexamethasone, 10 mM HEPES, and 1 nM insulin). The attached cells were then washed once in PBS and incubated overnight in low-glucose culture media (DMEM low glucose containing 10% FBS, 2% penicillin–streptomycin and 10 mM HEPES) for glucose production assays, or serum-free low-glucose culture medium (DMEM low glucose supplemented with 0.5% fatty acid free BSA, 2% penicillin–streptomycin, and 10 mM HEPES) for lipolysis assays. Following the overnight incubation, for glucose production assays, cells were washed twice with PBS and the media was replaced with 2 ml of substrate-replete glucose production medium (DMEM no-glucose base media containing 0.5% fatty acid-free BSA, 20 mM sodium lactate (50% 3-$^{13}$C), 2 mM sodium pyruvate, 2 mM GlutaMAX, MEM non-essential amino acids, and 10 mM HEPES). After a s6 hr incubation at 37 °C, the media was collected. Glucose concentrations were measured as described in the Biochemical Analysis section, and normalized to total protein measured using the bicinchoninic acid (BCA) assay. For lipolysis assays, cells were washed twice with PBS and the media was replaced with a fresh 2 ml volume of low-glucose culture medium. Cells were incubated at 37 °C for 6 hr, after which the media was collected and non-esterified fatty acid concentrations measured as described in the Biochemical Analysis section. The lipolysis rate was calculated after normalizing to total protein concentrations determined using the BCA assay. Flux ratios were measured using Equations 5-13 in *Supplementary file 2*, and back-calculated from net glucose production determined by measuring the glucose concentration in the media using the Sekisui Glucose Assay and assuming a linear rate of net glucose production during the 6 hr incubation.

## Glucose production in liver slices

Mice and rats were fasted overnight (16 hr) and sacrificed. Rodent livers were extracted and washed in Krebs-Henseleit buffer (KHB) containing 550 mM sodium chloride, 23 mM potassium chloride, 6.3 mM calcium dichloride, 10 mM magnesium sulfate, and 6.9 mM sodium phosphate monobasic. Rodent livers were cored into smaller bits using the Alabama R&D tissue coring press. Cored livers were sliced to the thickness of 230 microns, the lowest setting on the Alabama R&D tissue slicer. Liver slices were transferred to 24-well plates containing gluconeogenesis media (DMEM without glucose but with 20 mM lactate, 2 mM pyruvate, 10 mM HEPES, and 44 mM sodium bicarbonate) (400 ul for mouse livers and 500 ul for rat livers). The 24-well plates were placed in a tissue culture incubator (5% $CO_2$) and shaken at 80 RPM for 6 hr. At the end of the 6 hr incubation, liver slices and media were collected. Liver slice weights were measured on a scale. Glucose concentrations from liver slice media were measured using the Sekisui Glucose Assay. Glucose concentrations were normalized to liver weights measured in milligrams and microliters of gluconeogenesis media.

## In vitro oxygen consumption analysis

Primary hepatocytes were isolated from *ad lib* fed mice and rats by the Yale Liver Center's Cell Isolation core and plated recovery media as described previously *Camporez et al., 2013* in 24-well XF24 V7 cell culture plates coated with type I collagen. After 6–8 hr of recovery at 37 °C in 5% $CO_2$, cells were washed twice with PBS and the media was replaced with low-glucose culture media (DMEM base medium containing 5 mM glucose, 2 mM glutamine, and non-essential amino acids, pH 7.4), in which cells were cultured overnight. The next morning, as we have previously described (*Perry et al., 2020*), cells were washed twice with PBS and the media was replaced with 500 µL XF24 assay medium (DMEM base medium containing 5.5 mM glucose, 1 mM pyruvate, and 2 mM glutamine, pH 7.4) and equilibrated at 37 °C for 60 min. The Seahorse XFe 24 Analyzer was used to perform a mitochondrial stress test: after three baseline measurements of $O_2$ consumption (10 min apart), oligomycin (an inhibitor of ATP synthase) was injected, and three subsequent measures of $O_2$ consumption was performed using a 4 min mix/2 min wait/4 min measure protocol. Next, the uncoupler 2-[2-[4-(trifluoromethoxy) phenyl]hydrazinylidene]-propanedinitrile (FCCP) was injected to dissipate the proton gradient, with three $O_2$ consumption measurements taken as described above. Finally, rotenone (0.5 µM) and antimycin (10 µM) were injected to inhibit Complexes I and III, respectively. Oxygen consumption was normalized to total protein measured using the Pierce BCA Protein Assay.

## In vivo tracer analysis

Mice and rats received a 3 X (5 min) primed-continuous infusion of [3-$^{13}$C] sodium lactate (4.5 mg/kg body weight/min), [1, 2, 3, 4, 5, 6, 6-$^2$H$_7$] glucose (1.0 mg/kg/min), and [U-$^{13}$C$_{16}$] potassium palmitate (0.8 mg/kg/min) for 120 min. Tracers were infused into the jugular vein (mice) or the right atrium (rats) to ensure systemic delivery. After 100, 110, and 120 min, blood was collected from the tail vein (mice) or from the jugular venous catheter (rats) and centrifuged to obtain plasma, then animals were sacrificed with IV pentobarbital and their livers freeze-clamped in liquid nitrogen.

## Biochemical analysis

In plated hepatocyte and liver slice studies, glucose concentrations in the cell media were determined using the Sekisui Glucose Assay. Non-esterified fatty acid concentrations in the hepatocyte media were measured using the Sekisui Non-Esterified Fatty Acid kit.

Glucose production in hepatocytes was determined by measuring the glucose concentration in media after 1, 2, 3, 4, 5, and 6 hr of incubation. Glucose enrichment in hepatocytes, plasma, and livers was measured by gas chromatography/mass spectrometry (GC/MS). Samples were deprotonized using 1:1 barium hydroxide:zinc sulfate (100 μl for cells and 300 μl for ~100 mg liver samples, which were subsequently homogenized using a TissueLyser) and derivatized with 50 μl 1:1 acetic anhydride:pyridine. After 20 min heating to 65 °C, 50 μl methanol was added, and glucose enrichment ([$^{13}$C$_1$], [$^{13}$C$_2$] and, in plasma, [$^2$H$_7$] (all using the Chemical Ionization mode), [4, 5, 6-$^{13}$C$_1$] and [4, 5, 6-$^{13}$C$_2$] (both using the electron ionization mode)) were measured by GC/MS.

[$^{13}$C] alanine enrichment was measured by GC/MS (*Perry et al., 2016*). We have previously shown that flux through pyruvate kinase and malic enzyme – which would add [$^{13}$C] label to carbon 2 of alanine – is minimal under fasting conditions in normal rats *Perry et al., 2016*; therefore, the measured [$^{13}$C] alanine enrichment can be attributed entirely to [3-$^{13}$C] lactate enrichment. [$^{13}$C] malate enrichment was also measured by GC/MS (*Perry et al., 2017b*). Total malate enrichment and C1C2C3 malate enrichment were measured and the C2 + C3 malate enrichment was determined according to *Supplementary file 2*, equations 1 and 2, which relies on the assumption that the C4 enrichment is approximately equal to the C1 enrichment of malate (*Perry et al., 2017b*). Total and C4C5 glutamate enrichment was measured by LC-MS/MS (*Perry et al., 2018*). With $^{13}$C lactate infusion, no label enters glutamate carbon 5, so all label in the C4C5 fragment was assumed to be labeled in glutamate C4.

Plasma transaminase concentrations were measured by ELISA (Abcam; ALT: ab282882 [mouse]/ab234579 [rat], AST: ab263882 [mouse]/ab263883 [rat]). Liver glycogen concentrations were measured in 0 and 4 hr fasted mice and rats using the phenol-sulfuric acid method reported by *Schaubroeck et al., 2022*. PC activity (*Vatner et al., 2013*) and peroxidase activity (Sigma 'Enzymatic Assay of Peroxidase (EC 1.11.1.7)' protocol) were measured enzymatically in the liver.

## In vivo flux analysis

Endogenous palmitate and glucose production were determined using equation 3 in *Supplementary file 2*; *Perry et al., 2017a* comparing plasma enrichment to the infused tracer enrichment, measured using gas chromatography/mass spectrometry (GC/MS) as described in the Biochemical Analysis section. In the fasted/substrate-depleted and, therefore, glycogen-depleted state (*Perry et al., 2018*), endogenous glucose production can be attributed entirely to gluconeogenesis (equation 4). Based on equations previously described (*Perry et al., 2017b*), and after verifying minimal renal and hepatic bicarbonate enrichment by GC/MS (*Perry et al., 2020*), we measured the whole-body ratio of phosphoenolpyruvate carboxykinase (PEPCK) flux (i.e. gluconeogenesis from pyruvate) to total gluconeogenesis by mass isotopomer distribution analysis (equations 5 and 6), correcting for any [$^{13}$C$_2$] glucose synthesized from [$^{13}$C$_2$] trioses (equation 7). At steady state, V$_{PEPCK}$ is equal to the sum of pyruvate kinase flux and pyruvate carboxylase flux (V$_{PK}$ +V$_{PC}$); based on our previously published data, under fasting conditions pyruvate kinase flux is minimal, less than 10% of pyruvate carboxylase flux (*Perry et al., 2016*). Therefore, we can assume that the rate of gluconeogenesis from pyruvate (i.e. V$_{PEPCK}$) is approximately equal to V$_{PC}$.

Next, we measured the ratio of pyruvate carboxylase anaplerosis to citrate synthase flux (V$_{PC}$/V$_{CS}$) using the enrichment of liver alanine, malate, and glutamate (equation 8). This equation, in which pyruvate cycling is again assumed minimal, is derived in detail in our recent publication (*Perry et al., 2017b*). We then measured the fractional contribution of glycolytic carbons to the TCA cycle (i.e.

pyruvate dehydrogenase flux relative to citrate synthase flux, $V_{PDH}/V_{CS}$) (equation 9) using the model and assumptions we (*Befroy et al., 2014*; *Perry et al., 2018*; *Perry et al., 2016*; *Song et al., 2020*) and others *Alves et al., 2011*; *Petersen et al., 2016*; *Petersen et al., 2015* have described. Finally, absolute turnover rates (equations 10-13) were determined, utilizing the ratios measured with equations 5, 8, and 9, and the absolute gluconeogenesis rate measured using equations 3 and 4.

## Statistical considerations, power calculations, and statistical analysis

The sample size (n=4–6 in vitro replicates, 2–4 slices, or 4–6 animals in vivo) was calculated to supply 80% power at $\alpha$=0.05 to detect the expected twofold difference with 50% variance. Power calculations were performed using the ClinCalc online calculator. Groups in the in vivo and in vitro studies were compared using the two-tailed unpaired Student's t-test after confirming that the data met the assumptions of the test. The in vivo studies compared 4–6 biological replicates (unique animals), the liver slice studies compared 2–4 biological replicates, and the in vitro studies compared two biological replicates, with three technical replicates (separate wells) from each. No samples were excluded from analysis in the in vitro or ex vivo studies. Randomization and blinding were not possible during the in vivo studies because of the readily apparent differences between mice and rats, but all analyses were performed by investigators who were blinded as to species. No adverse events occurred. ARRIVE reporting guidelines were followed.

## Acknowledgements

The authors thank Traci LaMoia for her assistance in performing the liver slice experiments, Kathy Harry of the Yale Liver Center for isolating hepatocytes, and members of the Perry and Pollak labs for helpful discussions. This study was funded by grants from the U.S. Public Health Service (K99/R00 R00CA215315 [to R.J.P.], T32-GM0007324 [supporting N.D.A.], T32GM136651 [supporting B.P.L.], and P30DK034989 [supporting the Yale Liver Center]).

## Additional information

### Funding

| Funder | Grant reference number | Author |
|---|---|---|
| National Institutes of Health | R00CA215315 | Rachel J Perry |

The funders had no role in study design, data collection and interpretation, or the decision to submit the work for publication.

### Author contributions

Ngozi D Akingbesote, Brooks P Leitner, Conceptualization, Investigation, Writing – original draft, Writing – review and editing; Daniel G Jovin, Investigation, Writing – review and editing; Reina Desrouleaux, Dennis Owusu, Wanling Zhu, Zongyu Li, Investigation; Michael N Pollak, Conceptualization, Funding acquisition, Writing – review and editing; Rachel J Perry, Conceptualization, Supervision, Funding acquisition, Investigation, Writing – original draft, Writing – review and editing

### Author ORCIDs

Ngozi D Akingbesote http://orcid.org/0000-0001-6390-5729
Brooks P Leitner http://orcid.org/0000-0001-8744-412X
Daniel G Jovin http://orcid.org/0000-0002-8411-5942
Rachel J Perry http://orcid.org/0000-0003-0748-8064

### Ethics

Clinical trial registration NCT03738852.
Plasma samples were obtained, deidentified, from healthy, lean participants (all male, in order to maintain consistency with the rodent studies; age 26±3, body mass index 24.2±0.8; ethnicity data not collected) in a Yale University IRB-approved (protocol 0108012609) clinical study. All volunteers provided informed consent to participate in the study, and for the data to be published.

All animal studies were approved by the Yale University Institutional Animal Care and Use Committee (protocol 2019 and later 2022-20290).

## Decision letter and Author response

Decision letter https://doi.org/10.7554/eLife.78335.sa1
Author response https://doi.org/10.7554/eLife.78335.sa2

## Additional files

### Supplementary files
• Supplementary file 1. Primer sequences used for qPCR.

• Supplementary file 2. Flux ratios and absolute rates measured in mice infused with [3-$^{13}$C] lactate. APE indicates the atom percent enrichment (in animals infused with $^{13}$C tracer), TCA denotes the tricarboxylic acid cycle, and GNG denotes gluconeogenesis. By convention, $V_a$ represents the flux through pathway *a*.

• MDAR checklist

### Data availability

Source data generated for this manuscript are included in Source Data files 1-5. Source data for the gene expression analyses are in the original paper (DOI is provided in the manuscript: https://doi.org/10.1093/nar/gkg091). The source of the proteomics data is also provided in the manuscript: All liver raw mass spectrometry proteomics data were obtained from the Proteomics Identifications Database (PRIDE): mouse liver S9 fraction (PXD000733), rat liver S9 fraction (PXD000717) and human liver S9 fraction (PXD00734) from a previously published study (Golizeh et al., 2015).

The following previously published datasets were used:

| Author(s) | Year | Dataset title | Dataset URL | Database and Identifier |
|---|---|---|---|---|
| Brazma A, Parkinson H, Sarkans U, Shojatalab M, Vilo J, Abeygunawardena N, Holloway E, Kapushesky M, Kemmeren P, Lara GG, Oezcimen A, Rocca-Serra P, Sansone SA | 2003 | ArrayExpress | http://www.ebi.ac.uk/arrayexpress | ArrayExpress, E-MTAB-5166 |
| Golizeh M, Schneider C, Ohlund LB, Sleno L | 2015 | Proteomic Analysis of Mouse Liver S9 Fraction by 2D-LC-MSMS | https://www.ebi.ac.uk/pride/archive/projects/PXD000733 | PRIDE, PXD000733 |
| Golizeh M, Schneider C, Ohlund LB, Sleno L | 2015 | Proteomic Analysis of Rat Liver S9 Fraction by 2D-LC-MSMS | https://www.ebi.ac.uk/pride/archive/projects/PXD000717 | PRIDE, PXD000717 |
| Golizeh M, Schneider C, Ohlund LB, Sleno L | 2015 | Proteomic Analysis of Rat Liver S9 Fraction by 2D-LC-MSMS | https://www.ebi.ac.uk/pride/archive/projects/PXD00734 | PRIDE, PXD00734 |

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
