## [Editor Report]

Key metabolic processes have been shown to scale inversely with the body mass of different animals. This study provides direct evidence for metabolic scaling of key metabolic fluxes in the livers of mice and rats, as well as species-specific differences in the transcription and expression of enzymes involved in energy metabolism that could contribute to metabolic scaling. The finding suggests that metabolic scaling likely reflects multiple levels of regulation and have broad implications for studying animal metabolism and physiology.

---

## [Decision Letter]

**Decision letter after peer review:**

Thank you for submitting your article "Gene Expression and Tracer-Based Metabolic Flux Analysis Reveals Tissue-Specific Metabolic Scaling in vitro, ex vivo, and in vivo" for consideration by *eLife*. Your article has been reviewed by 3 peer reviewers, including Malcolm J McConville as the Reviewing Editor and Reviewer #1, and the evaluation has been overseen by David James as the Senior Editor.

Essential revisions:

The reviewers appreciated the new insights that the study provides into metabolic scaling with body mass and the use of different approaches to understand the mechanistic basis for scaling. However, a number of key issues were identified which need to be addressed.

1. All of the reviewers noted the need for further experimental data to support the conclusion that the species-specific differences in transcription of identified metabolic genes are reflected at the level of protein expression or enzyme activity. This could be addressed by reference to the author's own data, or to existing proteomics data-sets on the liver of different species and/or mRNA and proteome data within a single species.

2. Are there orthologues of the different liver enzymes that were found to scale with body size and are they also transcriptionally down-regulated as body size increases?

3. Do you see a similar transcriptional signature for other metabolic pathways that are known to inversely scale with body size (e.g. glycogen metabolism).

4. Specific sections in the manuscript (introduction, ideas/speculations) and some of the conclusions need to be changed/toned down in light of the reviewers' comments. Issues with the description of the flux analysis (reviewer 2) also need to be addressed to allow full assessment of the findings.

*Reviewer #1 (Recommendations for the authors):*

The title should be changed to reflect the lack of metabolic scaling in vitro assays.

Can the authors comment on the overall transcriptional activity of liver cells in different animals? Is it possible that global rates of transcription decrease in liver tissues of large animals and that the selective down-regulation of specific mRNA reflects differences in the rate of mRNA degradation?

Can the authors comment on levels of transcription or fluxes for other interconnected pathways of central carbon metabolism, such as the pentose phosphate pathway, glycogen turnover, (sugar) nucleotide biosynthesis? In particular, glycogen turnover has been shown to be much faster in mice than in humans (with cellular stores being exhausted during diurnal fasting in mice, but not in human tissues). It would be of interest to confirm that key enzymes in glycogen turnover exhibit scaling (at the level of transcription or activity) in the different animals and/or reference this example.

Can the authors comment on whether there are isoforms of the different enzymes that were found to scale with body size and whether the transcription of all isoforms was similarly related to body size?

It is proposed that enzymes involved in lipid peroxidation, and detoxification of hydrogen peroxide also inversely scale with body size. Is there any evidence for higher levels of oxidized lipids or hydrogen peroxide in mouse versus rat tissues (or tissues from other animals)?

The introduction is overly long and focused on summarizing the results of the current study.

The full names for all of the enzymes listed in Figure 1B (and the text) should be provided in the figure legend.

*Reviewer #2 (Recommendations for the authors):*

1. The title does not convey the findings well, since scaling was not observed in vitro.

2. To detect scaling, one needs to establish a linear relationship (in log-transformed variables). What is done in the current manuscript is that the ordering of trait values in terms of mass is identified; these are necessary but not sufficient for scaling. Specifications and clarifications are needed in multiple places in the manuscript.

3. The repetitive statements about correlation between mRNA expression and protein abundance are misleading and must be corroborated by data. The authors of the cited paper concluded "our data indicates that mRNA and protein levels correlate better than previously thought." This does not imply a strong correlation.

4. There is no support for allometric scaling (allometric would require substantially more data, see my point 2, above) claimed in Figure 1. Figure 1 can be supplemented with GO enrichment analysis to inspect how pervasive is the ordering pattern.

5. Broad spectrum of fluxes in line 160 is an overstatement. Identification of fluxes by MFA ala Wichert et al. would allow for estimation of fluxes on the level of a metabolic network. This is not addressed in the current work and should be discussed.

6. The section starting with line 226 is obsolete. Machine learning here refers to classical clustering and should have been named as such.

7. The section on ideas and speculations is far-fetched.

8. The analysis of gene expression should detail the one-way ANOVA, given that only two replicates were used. In addition, the statement about p-value adjustment on line 352 is not understandable as currently written.

9. The statement on line 78 is difficult to follow, some words are out of place.

10. Metabolic flux analysis is by definition tracer-based, so the adjective is not needed.

*Reviewer #3 (Recommendations for the authors):*

This work explores new parameters to characterize metabolic scaling. Instead of analyzing oxygen consumption, as has been used before, the authors focus on gene expression and metabolic fluxes. Gene expression data sets were analyzed for 5 species with widely varied body masses, whereas the metabolic studies were performed for mouse and rat, the closest phylogenetic species studied and the closest in body mass. A subset including an unspecified number of genes, predominantly related to metabolic pathways, followed the pattern of metabolic scaling across the 5 species. When the authors analyzed metabolic fluxes in vitro using primary hepatocytes, there was an absence of scaling, confirming similar reports. Studies ex vivo using liver slices (with limited replicates) and in vivo showed increased metabolic fluxes in mice compared with rats. Most of the measurements are focused on the liver, with one measurement of fatty acid turnover reflecting adipose tissue. This is at odds with the authors' apparent claim that there were differences in multiple tissues. Additionally, 11 of 16 metabolic enzymes that scaled also required molecular oxygen. Consequently, is not clear whether the gene expression scaling just reflects the differences in oxygen consumption, as has been reported.

Specific issues:

– The title implies that multiple tissues were analyzed, when in fact liver was the focus.

– How many genes from the total number of genes analyzed followed the pattern of metabolic scaling? Is this a pattern or particular for a set of the 20 genes shown in the manuscript? Are those genes expected to change scaling when mutated?

– Are the genes in Figure 1B. related to TCA cycle or electron transport chain?

– The authors should discuss the limitations of having performed metabolic flux studies only in the two most closely related species. Can we assume that scaling of metabolic fluxes continues over the very large range of body masses of the species that were not analyzed? Or is it possible that scaling only occurs up to a threshold value of body mass?

– The ex vivo studies appear to have been conducted using n = 2 for rats. Numbers of animals should be explicitly stated, as well as biological and technical replicates for in vitro and ex vivo experiments.

– Male animals were used. Is there a basis for not also studying female mice?

– The writing throughout the manuscript is laden with jargon and difficult to follow. There are sentences that are convoluted and do not make sense. Some of the jargon is not defined. The manuscript should be carefully edited throughout, including figure labeling and legends.

---

## [Author Response]

Essential revisions:The reviewers appreciated the new insights that the study provides into metabolic scaling with body mass and the use of different approaches to understand the mechanistic basis for scaling. However, a number of key issues were identified which need to be addressed.1. All of the reviewers noted the need for further experimental data to support the conclusion that the species-specific differences in transcription of identified metabolic genes are reflected at the level of protein expression or enzyme activity. This could be addressed by reference to the author's own data, or to existing proteomics data-sets on the liver of different species and/or mRNA and proteome data within a single species.

To address this important point, we have analyzed proteomics data from liver in mice, rats, and humans (unfortunately, all of the open-source data available lacked monkey and cow, so we were not able to extend the analyses to these species). Of the 20 genes that were found to correlate inversely with body mass in the transcriptomics analysis, eight translated proteins were found in the proteomics analysis. Of these eight, three exhibited an inverse correlation between body mass and protein expression.

Additionally, to begin to ascertain the generalizability of metabolic scaling at the protein level to tissues beyond liver, we also analyzed available proteomics in the left ventricle of the heart. One of the three proteins observed to correlate negatively with body mass in the proteomics analysis of liver, GLUL, was also shown to correlate negatively with body mass when its expression was assessed in the heart.

However, we recognize that we need to interpret these data carefully, because of the limited samples available: n=1 for each of these proteomics analyses, so statistical analysis was not possible. The sample size is limited because, unlike transcriptomics, it is not possible to combine proteomics data obtained from different instruments at different times under different processing conditions. We did not measure protein expression by Western blot because these results would require us to use antibodies with equal ability to bind to mouse and rat protein, which in most cases is infeasible. However, we have now measured activity of pyruvate carboxylase and of peroxidase in mouse and rat liver enzymatically. These data show lower activity of both enzymes in rats as compared to mice, suggesting that differences in enzyme activity as well as gene/protein transcription could underlie the metabolic scaling phenotype.

2. Are there orthologues of the different liver enzymes that were found to scale with body size and are they also transcriptionally down-regulated as body size increases?

To address this important point, we measured concentrations of the canonical liver enzymes, alanine aminotransferase (ALT) and aspartate aminotransferase (AST) in mice, rats, and humans. We do, indeed, observe scaling: both ALT and AST concentrations were lowest in humans, followed by rats, and highest in mice.

The transcriptional data reveal a reverse correlation with body mass in ALT (human gene name: *GPT*), but not AST, while the proteomic data did not demonstrate an inverse correlation with body mass in either enzyme. These data, as well as the lack of a reverse correlation between body mass and metabolic fluxes in hepatocytes, suggest that transcription/translation is one mechanism by which metabolic scaling may be accomplished in vivo, and that there are likely others as well (such as enzyme activity: data on pyruvate carboxylase and peroxidase activity have now been added in response to point #1).

3. Do you see a similar transcriptional signature for other metabolic pathways that are known to inversely scale with body size (e.g. glycogen metabolism).

The genes shown in Figure 2 are the only genes that exhibit a reverse correlation with body mass. We believe that while some of the phenomenon of metabolic scaling is directed by gene expression, some is not; this explains the dissociation between the lack of scaling of enzymes in the glycogen synthetic or glycogenolytic pathways, and the observed differences in rates of glycogenolysis. We have now added a direct assessment of the rate of net hepatic glycogenolysis between 0 and 4 hours of fasting, and confirm, in our hands, that glycogenolysis rates are lower in rats than mice, albeit with a smaller difference than other metabolic processes we measured in vivo.

We are not aware of other organ-specific metabolic processes that are well-established to scale with body size, after searching both prior to the initial submission, and upon preparing the revision.

4. Specific sections in the manuscript (introduction, ideas/speculations) and some of the conclusions need to be changed/toned down in light of the reviewers' comments. Issues with the description of the flux analysis (reviewer 2) also need to be addressed to allow full assessment of the findings.

We have substantially revised the introduction, including, but not limited to:

– Editing “Since at basal levels, gene expression of metabolic genes closely approximates protein levels (Schwanhäusser et al., 2011), we hypothesized that genes related to the regulation of oxygen consumption and substrate metabolism may scale, and thus provide a mechanistic basis for metabolic scaling” to “Beginning at the transcriptional level, we studied…”.

– Substantially reducing the description of both the transcriptomics and fluxomics studies.

– Editing the final sentence of the introduction to “Taken together, this study demonstrates variation of metabolic fluxes according to body size, extending prior studies of metabolic scaling, and provides unique insight into the regulation of metabolic flux across species.”

– Adding the words in red to “The concept of hierarchical regulation, whereby gene expression initiates the cascade that allows for flux through metabolic pathways (Rossell et al., 2005; Suarez and Moyes, 2012), provides a systems framework to begin to understand scaling.”

With regard to the ideas/speculations, we respectfully respond that we agree to an extent, but this section was written in response to an announcement from *eLife* on 11/9/2021 (*eLife* Latest: Including “Ideas and Speculation” in *eLife* papers | Inside *eLife* | *eLife* (elifesciences.org)), which appears to imply that this is the section in which such ideas are welcomed at *eLife* (emphasis added by our group):

“…As a result, the “Discussion” section of papers, which was once the home to much speculation, has over the years become a repository of only the most obvious and strongest conclusions, along with any accompanying caveats. And when authors try to include ideas that arise from, but which are not yet well supported by, the data, it is common for reviewers to demand that such speculation be removed.

While we agree that it makes sense to differentiate conclusions, we also feel something important is lost for both readers and authors by denying the people most familiar with the data the chance to freely speculate about its meaning and implications. Indeed it is one of the more common complaints of authors that current norms in peer review largely deny them this opportunity.

This section will be reviewed, but only for factual inaccuracies, clarity and speculation not germane to the paper. Some types of speculation – especially things that could be interpreted as suggestions for clinical practice – will not be allowed. But overall, this will be a place for authors to share their ideas about their work.”

As such, we believe that this section is written in accordance with the description of the section, in order to discuss “ideas that arise from, but which are not yet well supported by, the data.” We are completely open to editing if we misunderstood the purpose of this section, but we do respectfully believe that it is written consistently with the description of this section published in 2021.

*Reviewer #1 (Recommendations for the authors):*
The title should be changed to reflect the lack of metabolic scaling in vitro assays.

We agree, and have changed the title to “Gene and Protein Expression and Metabolic Flux Analysis Reveals Metabolic Scaling in Liver ex vivo and in vivo”.

Can the authors comment on the overall transcriptional activity of liver cells in different animals? Is it possible that global rates of transcription decrease in liver tissues of large animals and that the selective down-regulation of specific mRNA reflects differences in the rate of mRNA degradation?

It is possible that differences in the global rate of mRNA degradation between species contributes to differences in enzyme expression; however, our data showing that enzyme activity is different in mice versus rats would indirectly argue against that hypothesis.

However, to more directly address this point, we have now measured mRNA expression of three enzymes in different pathways that inversely correlate with body mass at the transcriptional level (*Glul, Lipe,* and *Dlst*) in livers from our own mice and rats relative to *Actb,* the gene that codes for b-actin. We find that expression of all three enzymes relative to *Actb* is lower in rats as compared to mice.

These data suggest that metabolic scaling on the transcriptional level is not a consequence of global alterations in mRNA turnover, because if, for instance, increased mRNA turnover were the cause of differential gene expression, we would expect no difference in the ratios of metabolic enzyme expression to b-actin expression.

Can the authors comment on levels of transcription or fluxes for other interconnected pathways of central carbon metabolism, such as the pentose phosphate pathway, glycogen turnover, (sugar) nucleotide biosynthesis? In particular, glycogen turnover has been shown to be much faster in mice than in humans (with cellular stores being exhausted during diurnal fasting in mice, but not in human tissues). It would be of interest to confirm that key enzymes in glycogen turnover exhibit scaling (at the level of transcription or activity) in the different animals and/or reference this example.

We did not observe scaling of enzymes in the glycogen synthesis or glycogenolysis pathways at the transcriptional or protein level (the enzymes shown in the transcriptomics analysis were the only metabolic enzymes that scaled). However, in this revision, we have now measured rates of net hepatic glycogenolysis. We do observe lower rates of net hepatic glycogenolysis in rats versus mice, albeit with a smaller difference than was observed in other fluxes measured in this manuscript.

Taken together, we believe that while some of the phenomenon of metabolic scaling is directed by gene expression, some is not; this explains the dissociation between the lack of an inverse relationship between body size and expression of enzymes in the glycogen synthetic or glycogenolytic pathways, and the observed differences in rates of glycogenolysis. We have added a comment in the Discussion clarifying that not all scaling can be explained at the transcriptional level:

“[I]t should be noted that metabolic scaling cannot fully be explained at the transcriptional level, because many rate-limiting enzymes in the metabolic processes measured in vivo did not scale at the transcriptional level, and only approximately half of genes that scaled at the level of mRNA scaled at the level of protein. Thus, it is likely that both transcriptional and other mechanisms – such as enzyme activity – are responsible for variations in metabolic flux per unit mass, inversely proportionally to body size.”

Can the authors comment on whether there are isoforms of the different enzymes that were found to scale with body size and whether the transcription of all isoforms was similarly related to body size?

Unfortunately, to our knowledge after an extensive search, the available liver transcriptomics and proteomics data do not allow us to assess potential scaling of different isoforms of key enzymes across species. However, this is an important possibility, and we have added a comment in the Discussion acknowledging this limitation:

“Additionally, the currently available data do not allow us to assess whether expression of certain isoforms of key metabolic enzymes scale differentially across species.”

It is proposed that enzymes involved in lipid peroxidation, and detoxification of hydrogen peroxide also inversely scale with body size. Is there any evidence for higher levels of oxidized lipids or hydrogen peroxide in mouse versus rat tissues (or tissues from other animals)?

We thank the reviewer for this excellent suggestion, and have now measured hepatic peroxidase activity in mouse and rat samples. We observe a 30% reduction in peroxidase activity in rats as compared to mice.

These data provide evidence for scaling in the peroxidation pathway, and have been added to the manuscript.

The introduction is overly long and focused on summarizing the results of the current study.

We acknowledge this issue and thank the reviewer for their feedback. In response, we have removed the following sections from the Introduction:

– …”substrate fluxes including lipolysis, gluconeogenesis, and substrate contributions to mitochondrial oxidation”.

– “Due to the relatively recent advances in high throughput mRNA sequencing and bioinformatics tools that allow for intra-specific data preprocessing (Bray et al., 2016; Conesa et al., 2016; Ritchie et al., 2015), we searched for a set of genes whose expression follows the pattern of metabolic scaling in the liver, the metabolic hub of mammals. Since at basal levels, gene expression of metabolic genes closely approximates protein levels (Schwanhäusser et al., 2011), we hypothesized that genes related to the regulation of oxygen consumption and substrate metabolism may scale, and thus provide a mechanistic basis for metabolic scaling.”

– “[Transcriptomics]…informed our isotope-tracer based in vitro and in vivo metabolic flux studies.”

– “[To determine if gene expression would reveal] a mechanistic read out of metabolic enzyme function,…”.

– [We utilized methods] to measure mitochondrial oxidation of glucose and fatty acids, gluconeogenesis from pyruvate and glycerol, and white adipose tissue lipolysis in rats and mice at multiple nodes of complexity: in vitro in hepatocytes; ex vivo in liver slices comprised of hepatocytes plus the surrounding stellate, Kupffer, and sinusoidal endothelial cells; and in vivo in conscious mice.”

– “gene expression in livers showed that scaling occurs to regulate oxygen consumption and substrate supply, isotope-based tracer studies in mice and rats demonstrated the mechanistic function of these enzymes in vivo which was only apparent in the living organism rather than plated cells.”

With these changes, the word count in the Introduction is reduced by 40% (from 704 to 422 words) and the section is substantially less focused on the results of the current manuscript.

The full names for all of the enzymes listed in Figure 1B (and the text) should be provided in the figure legend.

We thank the reviewer for this important suggestion and have now included the full names for all of the enzymes in Figure 1B (all of which are also contained in the text), as well as in Figure 2, in the respective figure legends.

Reviewer #2 (Recommendations for the authors):1. The title does not convey the findings well, since scaling was not observed in vitro.

We agree, and have changed the title to “Gene and Protein Expression and Metabolic Flux Analysis Reveals Metabolic Scaling in Liver ex vivo and in vivo”.

2. To detect scaling, one needs to establish a linear relationship (in log-transformed variables). What is done in the current manuscript is that the ordering of trait values in terms of mass is identified; these are necessary but not sufficient for scaling. Specifications and clarifications are needed in multiple places in the manuscript.

In several places in the current manuscript, we have replaced the term “metabolic scaling” with “metabolic trait ordering” or an equivalent term, including:

– In the Introduction, replacing “To determine whether flux through key implicated metabolic pathways scaled…” with “To determine whether flux through key metabolic pathways was ordered inversely to body size…”.

– Editing the phrase “while scaling of metabolic fluxes is not observed in the cell-autonomous setting, it is present in liver slices and in vivo” to replace “scaling” with “ordering”.

– Editing “Beginning at the transcriptional basis for metabolic scaling…” to “Beginning at the transcriptional level…” in the Introduction.

– Editing the comment in the Introduction that numerous genes “followed the pattern of metabolic scaling” to “were expressed at levels inverse to body size”.

– Reporting proteomics results in the Introduction: “Further analysis of liver proteomics revealed that approximately half of the genes in liver that were expressed inversely proportionally to body size at the transcriptional level, were also expressed at levels inversely proportional to body size at the level of protein expression”.

– Editing “Taken together, this study demonstrates systems regulation of metabolic scaling: gene expression in livers showed that scaling occurs to regulate oxygen consumption and substrate supply, isotope-based tracer studies in mice and rats demonstrated the mechanistic function of these enzymes in vivo which was only apparent in the living organism rather than plated cells. This study provides unique insight into metabolic scaling at the level of gene expression and metabolic enzyme function” to “Taken together, this study demonstrates systems regulation of the ordering of metabolic fluxes according to body size, and provides unique insight into the regulation of metabolic flux across species.”

– In a section moved from the Introduction to the Results: editing “we searched for a set of genes whose expression follows the pattern of metabolic scaling in the liver” to “we searched for a set of genes in the liver…whose expression correlates inversely with body mass”.

– In the Results, editing “we filtered out genes that followed the pattern of metabolic scaling. Genes that followed the pattern of mouse > rat > monkey > human > cow were predominantly…” to “we filtered out genes that followed the pattern of mouse > rat > monkey > human > cow. The genes that met these criteria were predominantly…”.

– Editing “In order to further understand the functional aspects of the metabolic genes that scale” to replace the word “scale” with “are expressed inversely proportionally to body size” in the Results.

– In the Results, editing “Genes from this scaling list” to remove the word “scaling”.

– In the Results, editing “[Genes] demonstrated a range of degrees of scaling,” to “[Genes] demonstrated a range of degrees of inverse correlation with body mass”.

– In the Figure 1 legend, replacing “(A) KEGG Pathway enrichment of all genes that follow pattern of allometric scaling…” with “(A) KEGG Pathway enrichment of all genes that are expressed with an inverse correlation to body mass”.

– In the Figure 1 legend, adding “To optimize use of the available space in the figure, the word “Scale” was used in the figure title to represent a pattern of inverse correlation between expression level and body size.”

– Replacing “scaling” with “the inverse relationship of expression of genes…” in the Results.

– Replacing “To understand whether or not certain genes that scale…” with “To understand whether or not certain genes that were expressed inversely proportionally to body size…” in the Results.

– In the Results, replacing “Eleven of sixteen critical metabolic enzymes that scaled” with “Eleven of sixteen critical metabolic enzymes that were expressed inversely proportionally to body size”.

– Adding “To examine the possibility that the inverse correlation between body mass and gene expression observed in the transcriptomics analysis could be a consequence of global alterations in mRNA…” in the Results.

– Adding “We found that all three enzymes (*Glul*, *Lipe*, and *Dlst*) scaled relative to b-actin (Figure 2—figure supplement 1A-C), whereas structural genes (collagenase 3 [*Mmp3*] and *Larp1*) did not (Figure 2—figure supplement 1D-E), indicating that the differences in metabolic gene expression observed across species…” in the Results.

– In the Results, replacing “Scaling of genes that are associated with interorgan crosstalk provide evidence for mechanisms of scaling in vivo, that would not be apparent in plated cells” with “The inverse correlation between body size and expression of genes that are associated with interorgan crosstalk is consistent with scaling in vivo that would not be apparent in plated cells”.

– In the Results, replacing “…and *Dlst* of the TCA cycle also scaled (Figure 2D-E)” with “…and *Dlst* of the TCA cycle also correlated inversely with body size (Figure 2D-E)”.

– Replacing “Metabolic genes that scale” with “Metabolic genes that are expressed inversely proportionally to body size” in the Results.

– In the Results, describing the alterations in transaminase concentrations as “consistent with scaling at the level of protein expression,” not demonstrating scaling.

– Similarly, in the Results, describing differences in metabolic enzyme activity as “suggesting scaling at the level of metabolic enzyme activity,” not demonstrating it.

– Editing “we asked whether glucose production would scale” to “we asked whether glucose production would be different” in the Results.

– Removing (in the description of the liver slice glucose production data) “demonstrating metabolic scaling of gluconeogenesis” in the Results.

– In the Results, editing “This conservation of differences in metabolic scaling applied not only to…” to “This conservation of differences in metabolic fluxes applied not only to…”.

– Replacing “demonstrating coordination of systemic metabolic scaling” to “demonstrating coordination of systemic metabolism” in the Results.

– Editing the Figure 5 legend from “A comprehensive analysis of systemic metabolic fluxes reveals in vivo metabolic scaling” To “A comprehensive analysis of systemic metabolic fluxes suggests in vivo metabolic scaling”.

– Replacing “metabolic scaling” with “flux” in the description of Figure 6 (Results).

– Replacing “metabolic scaling” with “metabolic differences” in the Figure 6 legend.

– Replacing “demonstrating” with “suggesting” in the phrase “demonstrating systemic conservation of metabolic scaling” in the Discussion.

– Editing “those genes for which scaling is observed” to “those genes for which an inverse relationship of expression with body mass is observed” in the Discussion.

– In the Discussion, replacing “the collection of genes that scale” with “the collection of genes whose expression is inversely correlated with body mass”.

– Editing “It is also informative to consider the settings in which metabolic scaling was not observed” to “It is also informative to consider the settings in which we did not observe alterations in metabolic flux” in the Discussion.

– Replacing “demonstrating systemic conservation of metabolic scaling” to “suggesting systemic conservation of the inverse relationship between metabolic flux and body mass” in the Discussion.

– In the Discussion, replacing “…in vivo observations (where livers exhibited robust metabolic scaling)” with “in vivo observations (where all fluxes proceeded faster in mice than in rats)”.

– In the Discussion, modifying “…[these data] emphasize the need to employ tracer methods in vivo to generate a comprehensive picture of the physiologic role of metabolic scaling” to “[these data] emphasize the need to employ tracer methods in vivo to generate a comprehensive picture of the physiologic role of differences in metabolic fluxes between species”.

– In the Discussion, modifying “This study sought to address this issue by examining the generalizability of metabolic scaling” to “This study sought to address this issue by examining the generalizability of the inverse relationship between body mass and metabolic rates.”

– Of note, we did not edit the section of the Introduction or Discussion in which we discussed prior literature in which the term “metabolic scaling” was used, generally referring to data demonstrating that metabolic rates (i.e. oxygen consumption) scale to ¾ of an organism’s body mass.

In addition, we would like to respectfully suggest that the definition of metabolic scaling defined as the relationship of metabolic flux to ¾ of body mass is defined because of the input data, while our work is asking a different question: we are not suggesting that oxygen consumption per whole organism does not scale inversely to ¾ of body mass; however, the fact that organ-specific metabolism may scale differently does not mean that the observed phenotype is not metabolic scaling. It is possible that different metabolic processes and/or metabolism in different organs scale differently, and in fact, this is what our data suggest. To address this important point, we have added a comment in the Discussion:

“It is important to note that the metabolic processes which we observed to be higher in mice as compared to rats did not necessarily adhere quantitatively to the classic metabolic scaling relationship, with metabolic rates proportional to three-quarters of body mass. This speaks to the fact that the scaling relationship is multidimensional: it is entirely conceivable that whole-body oxygen consumption could be proportional to three-quarters of body mass, while other metabolic processes may exhibit a different scaling relationship. Further studies across species beyond rodents will be required to address this question.”

3. The repetitive statements about correlation between mRNA expression and protein abundance are misleading and must be corroborated by data. The authors of the cited paper concluded "our data indicates that mRNA and protein levels correlate better than previously thought." This does not imply a strong correlation.

We thank the reviewer for raising this important point, and have now added proteomics data, as described in our response to Reviewer 2’s Public Evaluation comment #1. In addition, in response to Reviewer 2’s point, we have softened the discussion of the implications of the transcriptomics data as described in our response to Reviewer #2’s point 1 (our response begins “Additionally, we have substantially softened the description of the implications of the transcriptomics data in the Introduction and Discussion…”).

4. There is no support for allometric scaling (allometric would require substantially more data, see my point 2, above) claimed in Figure 1. Figure 1 can be supplemented with GO enrichment analysis to inspect how pervasive is the ordering pattern.

Respectfully, we reiterate our response to the previous comment to emphasize that it is our view that scaling may occur across different scales when examining different metabolic processes, and while oxygen consumption has been the setting of metabolic scaling that has been most studied, this does not mean that the concept of metabolic scaling must be defined exclusively as the relationship between body mass and oxygen consumption.

5. Broad spectrum of fluxes in line 160 is an overstatement. Identification of fluxes by MFA ala Wichert et al. would allow for estimation of fluxes on the level of a metabolic network. This is not addressed in the current work and should be discussed.

We have removed the phrase “broad spectrum of fluxes”, and fully agree that the metabolic flux analysis discussed conceptually by Wiechert (Metabolic Engineering 2001) represents a means of utilizing stable isotope tracer analysis to generate a comprehensive picture of metabolic fluxes. We fully agree that the conceptual framework laid forth by Dr. Wiechert presents a means of comprehensively assessing substrate metabolism. However, we would respectfully submit that this is a theoretical paper presenting no data in live animals. We completely agree that such an analysis would be of great interest in further widening the scope of our understanding of metabolic scaling and have added a comment to this effect in the Discussion; however, we believe that this is beyond the scope of the current manuscript: “Future studies using metabolic flux analysis may have further capacity to generate new insights as to the scope and mechanism of metabolic scaling (Wiechert, 2001).”

6. The section starting with line 226 is obsolete. Machine learning here refers to classical clustering and should have been named as such.

We have replaced “machine learning” with “classical clustering” in both the section title in the Results, and in the description. We thank the reviewer for this helpful comment.

7. The section on ideas and speculations is far-fetched.

Much of our response here is duplicated from our response to the editor’s summary of the points which must be addressed in the revision. We respectfully respond that we agree to an extent, but this section was written in response to an announcement on 11/9/2021 (*eLife* Latest: Including “Ideas and Speculation” in *eLife* papers | Inside *eLife* | *eLife* (elifesciences.org)), which appears to imply that this is the section in which such ideas are welcomed at *eLife*:

“…As a result, the “Discussion” section of papers, which was once the home to much speculation, has over the years become a repository of only the most obvious and strongest conclusions, along with any accompanying caveats. And when authors try to include ideas that arise from, but which are not yet well supported by, the data, it is common for reviewers to demand that such speculation be removed.

While we agree that it makes sense to differentiate conclusions, we also feel something important is lost for both readers and authors by denying the people most familiar with the data the chance to freely speculate about its meaning and implications. Indeed it is one of the more common complaints of authors that current norms in peer review largely deny them this opportunity.

This section will be reviewed, but only for factual inaccuracies, clarity and speculation not germane to the paper. Some types of speculation – especially things that could be interpreted as suggestions for clinical practice – will not be allowed. But overall, this will be a place for authors to share their ideas about their work.”

As such, we believe that this section is written in accordance with the description of the section, in order to discuss “ideas that arise from, but which are not yet well supported by, the data.”

8. The analysis of gene expression should detail the one-way ANOVA, given that only two replicates were used. In addition, the statement about p-value adjustment on line 352 is not understandable as currently written.

The criteria for one-way ANOVA test were met, such that (1) the samples are independent, (2) each sample is from a [presumably] normal distributed population per species, and (3) assumption of homoscedasticity across each group. A reference to the use of the specific statistical test used can be found here: https://docs.scipy.org/doc/scipy/reference/generated/scipy.stats.f_oneway.html#r74f03ee7d776-2

The text has now been adjusted to reflect that an α level of 0.05 was maintained for statistical significance across multiple statistics. A Bonferroni correction was applied in order to obtain a p-value of 0.01 for multiple comparisons for this analysis:

“To assess for differences in gene expression in genes displayed on the metabolic map, a one-way ANOVA with a Bonferroni correction for multiple comparisons (thus we used an adjusted p-value threshold for this test of 0.01, corresponding to an α level of 0.05, for significance) using the Python Scipy package (Virtanen et al., 2020).”

9. The statement on line 78 is difficult to follow, some words are out of place.

We apologize for any confusion that our writing generated. This has now been clarified as: “To determine if gene and protein expression would correlate with enzyme activity and metabolic flux, we performed a comprehensive assessment of liver metabolism in vivo and in vitro…”

10. Metabolic flux analysis is by definition tracer-based, so the adjective is not needed.

We agree, although there have been many unfortunate examples of metabolomics data being improperly interpreted as equivalent to flux, and we aimed to make clear that this was not such a case. Nevertheless, the reviewer is without question correct, and we have removed “tracer-based” from the title and the abstract.

Reviewer #3 (Recommendations for the authors):This work explores new parameters to characterize metabolic scaling. Instead of analyzing oxygen consumption, as has been used before, the authors focus on gene expression and metabolic fluxes. Gene expression data sets were analyzed for 5 species with widely varied body masses, whereas the metabolic studies were performed for mouse and rat, the closest phylogenetic species studied and the closest in body mass. A subset including an unspecified number of genes, predominantly related to metabolic pathways, followed the pattern of metabolic scaling across the 5 species. When the authors analyzed metabolic fluxes in vitro using primary hepatocytes, there was an absence of scaling, confirming similar reports. Studies ex vivo using liver slices (with limited replicates) and in vivo showed increased metabolic fluxes in mice compared with rats. Most of the measurements are focused on the liver, with one measurement of fatty acid turnover reflecting adipose tissue. This is at odds with the authors' apparent claim that there were differences in multiple tissues. Additionally, 11 of 16 metabolic enzymes that scaled also required molecular oxygen. Consequently, is not clear whether the gene expression scaling just reflects the differences in oxygen consumption, as has been reported.

We thank the reviewer for raising this interesting point, as it is possible: as the reviewer as mentioned, oxygen consumption has been shown to exert numerous metabolic alterations on cells. We have now added a comment acknowledging this point in the Discussion:

“[O]ur data do not allow us to ascertain whether differences in oxygen consumption orchestrate metabolic alterations, as has been suggested in the setting of cancer cells (Nakazawa et al., 2016), or metabolic alterations require changes in oxygen consumption.”

Specific issues:– The title implies that multiple tissues were analyzed, when in fact liver was the focus.

Liver was the focus, as the reviewer says, but white adipose tissue was also assessed in vivo with the inclusion of whole-body lipolysis. We have now added transcriptomic data from heart as well. However, we completely agree with the reviewer’s point that the primary focus of this manuscript was and is liver metabolism. As such we have edited the title to “Gene and Protein Expression and Tracer-Based Metabolic Flux Analysis Reveals Metabolic Scaling in Liver ex vivo and in vivo”.

– How many genes from the total number of genes analyzed followed the pattern of metabolic scaling? Is this a pattern or particular for a set of the 20 genes shown in the manuscript? Are those genes expected to change scaling when mutated?

The genes shown were the only ones to show an inverse correlation between gene expression and body mass. We interpret this as indicating that some of the inverse correlation between body mass and metabolic rate might be explained by variations in gene transcription (and even less by protein translation, as evidenced by the fact that only three of eight genes that scaled at the level of transcription and were contained in the proteomics dataset showed an inverse correlation with body size, although we must be clear that the proteomics analysis contained a limited n of 1). However, there must also be mechanistic underpinnings of the metabolic scaling phenotype that are outside of gene/protein expression, such as enzyme activity, as evidenced by our liver pyruvate carboxylase and peroxidase activity data.

To our knowledge, there are no data to suggest whether or not flux through the metabolic enzymes identified through the transcriptomics analysis would change when mutated. However, one would fully expect that, for instance, rendering these enzymes constitutively active would increase flux through those pathways, and might contribute to an overall increase in oxygen consumption. We would anticipate that if, for example, an enzyme in a rat were activated by a mutation to increase its activity to that observed in a mouse, that the metabolic scaling phenomenon would disappear. Future studies will be necessary to dissect this possibility. We have added a comment to this effect in the final paragraph of the Discussion:

“It may be that if one were to alter the activity of an enzyme in a larger species using a mutation to increase it to the level of activity of the same enzyme in a smaller species, that the phenomenon of metabolic scaling might disappear. Future work will be required to examine this possibility.”

– Are the genes in Figure 1B. related to TCA cycle or electron transport chain?

We thank Reviewer #3 for raising this point, and fully agree that a more accurate term is “electron transport." Thus, we have edited Figure 1B accordingly:

– The authors should discuss the limitations of having performed metabolic flux studies only in the two most closely related species. Can we assume that scaling of metabolic fluxes continues over the very large range of body masses of the species that were not analyzed? Or is it possible that scaling only occurs up to a threshold value of body mass?

Based on the existing data and computational modeling, it is likely that there is not such a threshold, at least in mammals, although our study cannot answer that question. We have, as suggested, added a section to address this important point:

“Additionally, there are limitations to the fact that metabolic flux studies were performed only in the two arguably most related species included in the transcriptomics analysis (with monkey and human perhaps being similarly related). Our laboratory does not have the capacity to perform flux analysis in larger or smaller species, but we fully recognize that widening our scope beyond the 10-fold range in body size between mice and rats may yield different results. For example, it is possible that the mathematical relationship between body size and metabolic flux may reach a threshold of body size, after which this relationship may change. However, experimental and computational data suggest that the metabolic scaling relationship with regard to oxygen consumption holds at least from birds to cattle, and that a similar – but not identical – scaling relationship exists in plants (Niklas and Kutschera, 2015), so we posit that it is likely that if such a threshold exists, it occurs at a body size orders of magnitude larger than the rodents included in our study.”

– The ex vivo studies appear to have been conducted using n = 2 for rats. Numbers of animals should be explicitly stated, as well as biological and technical replicates for in vitro and ex vivo experiments.

We have now added the numbers for each experiment (biological and, if applicable, technical replicates) to the corresponding figure legends.

– Male animals were used. Is there a basis for not also studying female mice?

We agree that potential sex differences in metabolic scaling are an important point, and as such have added analyses of metabolic fluxes in male (duplicated from an earlier figure, as noted in the figure legend) and female rats and mice. We did not observe any differences in glucose production, pyruvate carboxylase flux, citrate synthase flux, the contribution of glucose or fatty acids to total mitochondrial oxidation, or whole-body lipolysis (fatty acid turnover) between male and female mice or male and female rats: all fluxes were similarly lower in rats as compared to mice.

– The writing throughout the manuscript is laden with jargon and difficult to follow. There are sentences that are convoluted and do not make sense. Some of the jargon is not defined. The manuscript should be carefully edited throughout, including figure labeling and legends.

We have edited extensively in response to this important comment, and hope the reviewer will agree that the manuscript is much clearer after revision.